# Warming of hot extremes alleviated by expanding irrigation

Wim Thiery [1,2]*, Auke J. Visser [3], Erich M. Fischer [1], Mathias Hauser [1], Annette L. Hirsch [4], David M. Lawrence [5], Quentin Lejeune [6], Edouard L. Davin [1] & Sonia I. Seneviratne [1]

Irrigation affects climate conditions – and especially hot extremes – in various regions across the globe. Yet how these climatic effects compare to other anthropogenic forcings is largely unknown. Here we provide observational and model evidence that expanding irrigation has dampened historical anthropogenic warming during hot days, with particularly strong effects over South Asia. We show that irrigation expansion can explain the negative correlation between global observed changes in daytime summer temperatures and present-day irrigation extent. While global warming increases the likelihood of hot extremes almost globally, irrigation can regionally cancel or even reverse the effects of all other forcings combined. Around one billion people (0.79–1.29) currently benefit from this dampened increase in hot extremes because irrigation massively expanded throughout the 20$^{th}$ century. Our results therefore highlight that irrigation substantially reduced human exposure to warming of hot extremes but question whether this benefit will continue towards the future.

[1] Institute for Atmospheric and Climate Science, ETH Zurich, Universitaetsstrasse 16, 8092 Zurich, Switzerland. [2] Department of Hydrology and Hydraulic Engineering, Vrije Universiteit Brussel, Pleinlaan 2, 1050 Brussels, Belgium. [3] Meteorology and Air Quality group, Wageningen University, Droevendaalsesteeg 3a, 6708PB Wageningen, the Netherlands. [4] ARC Centre of Excellence for Climate Extremes, University of New South Wales, 2052 Sydney, Australia. [5] National Center for Atmospheric Research, Boulder, CO, USA. [6] Climate Analytics, Ritterstrasse 3, 10969 Berlin, Germany. *email: wim.thiery@vub.be

rrigation is one of the land management practices with the largest biogeophysical effect on climate[1]. Imprints of irrigation on local climates have been detected from in-situ temperature and energy flux observations[2–4] as well as in remotely sensed soil moisture datasets[5,6]. Several independent climate modelling studies have moreover shown that intense irrigation in the Indian subcontinent can delay the onset of the Indian Summer Monsoon[7–10], but also influences precipitation patterns in areas away from irrigation hotspots[9–12]. While the influence of irrigation on annual mean temperatures remains limited, emerging evidence reveals a large impact of irrigation on temperature extremes, with a particularly strong cooling during the hottest days of the year[2,3,10,13,14].

Despite this accumulating evidence, irrigation—and land management in general—have historically not been considered in assessments of drivers of observed climate change[15,16]. This is remarkable, since the irrigated area massively expanded from 0.63 million km$^2$ in 1900 to 3.06 million km$^2$ in 2005[17,18], thereby covering an area about the same size as India.

In this study, we use observational data and global climate simulations to isolate the climatic effects of irrigation from the warming induced by all other combined forcings, the latter dominated by anthropogenic greenhouse gas emissions[15]. We reconstruct the contribution of irrigation expansion to the observed change in average daily maximum temperature during the hottest month of the year (TXm) by applying a recently developed window searching algorithm[19,20] to the global gridded CRU temperature dataset (see 'Methods'). A major advantage of this method is that it can isolate an individual forcing from a lumped signal based on information from the surrounding area, which means that it can be directly applied to observations and multi-forcing climate simulations. We then perform a number of earth system model experiments that enable the comparison of irrigation effects on climate against other anthropogenic forcings and that can be analysed on daily instead of monthly time scales (see 'Methods'). Observational and model results consistently highlight a strong irrigation-induced cooling effect during warm extremes in intensely irrigated regions. In some regions, irrigation expansion cancels or even reverses the effects of all other forcings combined.

## Results

**Observational analysis**. We find that daytime temperatures during the hottest month of the year (TXm) have warmed less during the 20th century over areas with substantial irrigation expansion (>25% of grid cell area; Fig. 1a). This effect is strongest over Pakistan, India, Nepal and Bangladesh (Fig. 1b, hereafter referred to as South Asia), a global hotspot of irrigation activity (Supplementary Fig. 1). While constituting only a small portion of the global land area (0.5%), pixels with an increase in irrigated grid cell fraction above 35% generally experienced a cooling trend. These results suggest that irrigation has played an important role in the evolution of high temperatures on land.

Results of applying the reconstruction method indicate that irrigation expansion has a cooling influence on TXm which increases with greater irrigation extent (Fig. 1c, d). Subtracting the irrigation-induced cooling ($\Delta$TXm$_{irr}$) from the total change ($\Delta$TXm) indeed largely removes the initially observed negative relationship (Supplementary Fig. 2), confirming that much of the cooling of TXm in the observational record can be explained by irrigation expansion throughout the 20th century.

**Earth system modelling**. Since global-gridded observations either omit daily extremes or lack spatio-temporal coverage in irrigation hotspots (see Supplementary Note 1), we turn to earth system modelling for investigating irrigation influences on the occurrence of daily temperature extremes (TX) rather than the intensity of monthly hot extremes (TXm). We perform targeted model experiments with the Community Earth System Model (CESM) to isolate the effects of irrigation expansion from the global warming signal (see 'Methods'), and analyse hot extremes on daily instead of monthly time scales. Although the irrigation-induced cooling in monthly temperature extremes is larger in CESM than in the observation-based estimate, the two lines of evidence yield consistent results (Fig. 1, Supplementary Fig. 3) and are corroborated by the comparison of simulated and satellite-based land surface temperatures showing consistent and substantial irrigation-induced cooling in intensely irrigated regions (Supplementary Note 2, Supplementary Figs. 4–6). These analyses reaffirm earlier conclusions[10] that the model can be used to study the effects of irrigation on climate extremes, while the exact magnitude of the effect needs to be interpreted with caution. To compare the effect of irrigation to the anthropogenic warming signal, we use the well-established probability ratio (PR) metric[21,22], which characterises the factor by which the probability of an event has changed under a given forcing (see 'Methods').

Nearly everywhere across the globe, global warming has increased the likelihood of hot extremes, here defined as the local 99th TX percentile in the early 20th century (that is, the daytime temperature expected on average once every 100 days) (Fig. 2a). Simultaneously, expanding irrigation led to a reduction of the likelihood of these hot extremes (Fig. 2b). Unlike the global warming signal, this reduction is mostly limited to present-day irrigation hotspots (Supplementary Fig. 1), with only the Sahara as a notable exception (Fig. 2b). Over South Asia, irrigation locally reduced the likelihood of hot extremes by a factor of 2–8, with particularly strong effects over the Indo-Gangetic Plain. Combining these two competing effects highlights that in some regions, irrigation partly or completely reverses the anthropogenic warming of hot extremes (Fig. 2c). Consequently, irrigation expansion has prevented these regions from experiencing strong(er) warming rates during hot days.

The median PR from global warming is between 2 and 3 for the 99th percentile TX across various regions (Supplementary Fig. 7). This implies that, without irrigation, hot days occurring 3–4 times a year in the early 20th century would now be expected 6–10 times a year. The global warming-induced PR moreover increases for higher percentiles (Fig. 3), meaning that frequency changes are larger for high-end extremes. This partly follows from the definition of the metric: the rarer the event in the reference case, the larger the potential for strong relative changes (see Eq. 3)[22]. In addition, however, extremes are warming consistently faster compared with mean temperatures[23], which has previously been attributed to increasing sensitivity to soil desiccation in the tail of the temperature distribution[24,25]. When considering total global land area, irrigation expansion had limited influence on global median PR values (Fig. 3a). However, across irrigated lands, as well as over entire South Asia, it resulted in a substantial reduction in the likelihood of hot extremes (Fig. 3b, c). Similar to the global warming signal, the more intense the definition of the extreme the greater the reduction due to irrigation expansion. This can be understood from the nature of irrigation and land-climate dynamics: first, more water is applied during hot days (Supplementary Fig. 8) as their occurrence typically coincides with crop growing seasons and in many regions also with precipitation deficits[26]. Second, irrigation will reduce the land-atmosphere coupling strength of a given region and thereby its sensitivity to temperature variability[24] (Supplementary Fig. 9; Supplementary Note 3). Combining the effects of global warming and irrigation suggests that there has been virtually no change in

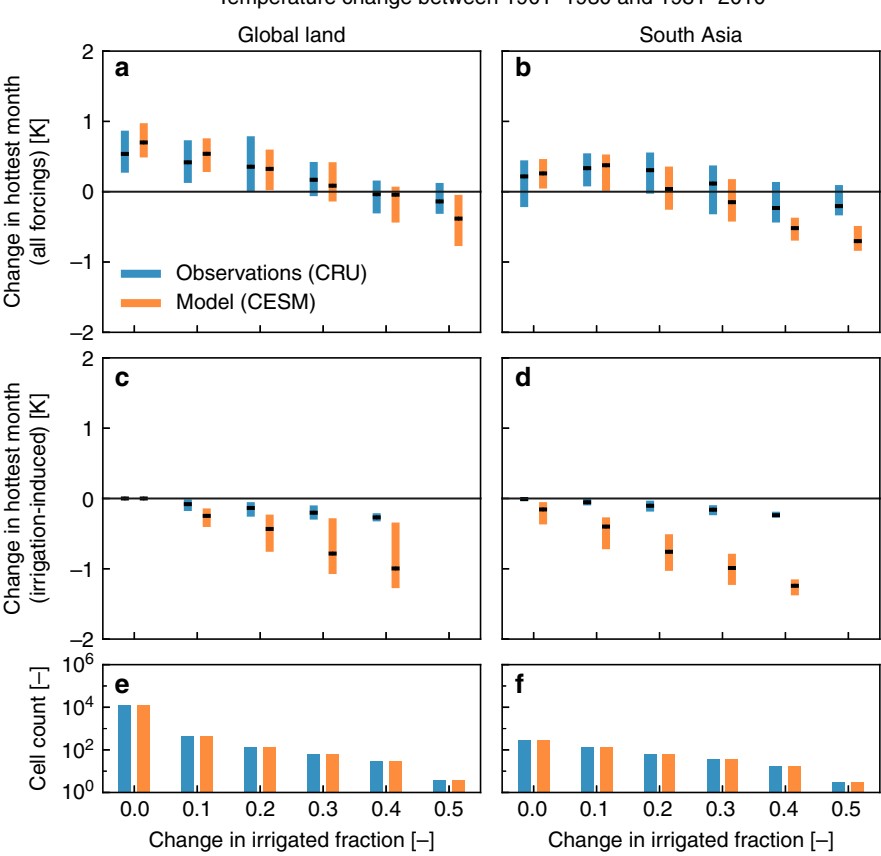

**Fig. 1 Observed warming rates affected by irrigation.** Boxplots of the total ($\Delta TXm$, **a**, **b**) and irrigation-induced ($\Delta TXm_{irr}$, **c**, **d**) change in average daily maximum temperature during the hottest month of the year for global land (**a**, **c**) and South Asia (**b**, **d**) between 1901 and 1930 and 1981 and 2010. Results were binned by the change in grid cell fraction equipped for irrigation (hereafter referred to as irrigated fraction $f_{irr}$) between both periods. Cell counts per bin for $\Delta TXm_{irr}$ are indicated in **e**, **f**. Blue bars represent results from the CRU observational data, orange bars show results from the CESM global climate simulations. Boxplots indicate the spatial distribution (centre line: median; box limits: upper and lower quartiles; whiskers and outliers: not shown) and are only plotted for bins containing ≥5 pixels. Note that most grid cells with $f_{irr}>0.5$ for the global land are situated in South Asia.

the likelihood of hot extremes across South Asia and irrigated lands (Fig. 3b, c). In other regions of the world these effects lead to a reduction in the overall PR (Supplementary Fig. 7). Our findings are robust even when accounting for spatial variability by considering spatial boxplots instead of medians (Supplementary Fig. 10). Moreover, excluding irrigation from the early 20th century control simulation—thereby increasing the relative global warming effect—only strengthens the main conclusions of our analysis (Supplementary Fig. 11). However, we note that our results only refer to the forced response, that is, we average across a range of ensemble members to reduce the effect of natural variability, whereas the actual observed frequency change at the local to regional scale is additionally influenced by natural variability superimposed on the forced pattern.

## Discussion

Disentangling individual contributions to observed temperature changes is needed to unravel the historic influence of anthropogenic greenhouse gas emissions and to improve the reliability of future climate projections. It has for instance been reported that historical deforestation increased the likelihood of hot extremes in northern mid-latitudes in a set of observationally constrained Earth system models[20,27]. This is a priori not incompatible with the cooling effect of irrigation which is found mainly in South Asia, but future work is needed to formally attribute observed temperature changes[2–4,16] to individual land

cover and land management changes using a consistent multi-model framework[28–30].

While conducted with a state-of-the-art Earth system model and with great care of achieving realism, our simulations remain characterised by a number of limitations. To start, we assume fixed irrigation extent (of the year 1915 and 2000, respectively) in our otherwise transient simulations as the current model version cannot handle transient irrigation area. Furthermore, we consider only one crop type (generic C3 crops) and we do not account for variations in irrigation water sources (such as groundwater pumping or rainwater tanks[31,32]), and irrigation techniques (such as sprinkler, drip, flood or ponding irrigation[33–38]). Instead CESM extracts required irrigation amounts from surface runoff and applies it directly to the soil surface, where it can either infiltrate, evaporate or runoff. We thus ignore potential local water availability limitations, but note that regional patterns of applied irrigation amounts are realistic when compared with census data[10]. This is confirmed by a set of land-only sensitivity experiments showing that the default irrigation parameter settings are most suited for representing irrigation quantities in South Asia (see Supplementary Fig. 12 and Supplementary Note 4). However, even though our simulated irrigation quantities closely match observed values over South Asia[10], water ponding is currently disabled in CESM, which likely leads to an underestimation of irrigation impacts on temperature in South, East, and Southeast Asia where paddy fields are widespread[36]. Comparison of simulated and satellite-based land surface

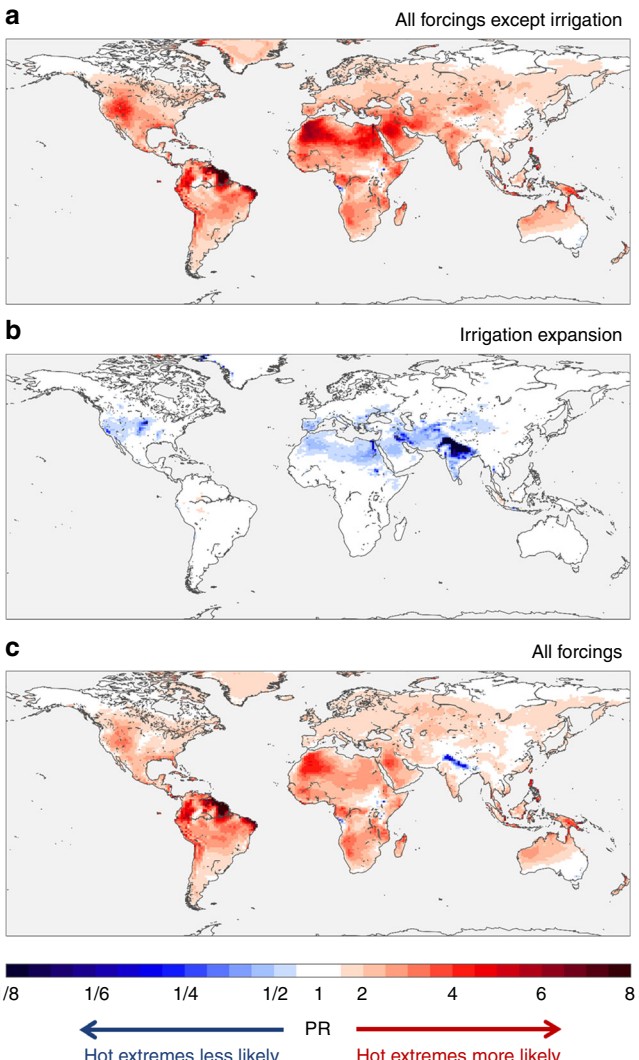

**a** All forcings except irrigation

**b** Irrigation expansion

**c** All forcings

1/8 1/6 1/4 1/2 1 2 4 6 8

PR

← Hot extremes less likely Hot extremes more likely →

**Fig. 2 Change in probability of hot extremes from expanding irrigation and other forcings.** Ensemble-mean likelihood of exceeding 99th percentile of daytime temperature (TX) as simulated by CESM, considering all forcings except irrigation (**a**), irrigation expansion only (**b**), and all forcings including irrigation expansion (**c**). Probability ratios (PRs) are shown for the present-day (1981–2010) relative to the early 20th century reference period (1901–1930), except for **b** where the reference is a counter-factual present-day world without irrigation.

temperatures indeed confirms that CESM appears to underestimate irrigation effects on present-day land surface temperatures across South Asia (see Supplementary Note 2). Implementing paddy irrigation in CESM would thus be beneficial, especially since observational studies[39–41] demonstrate that paddy field expansion in China may locally lead to land surface temperature reductions on the order of 1–2 K on average during the growing season.

In summary, we showed that irrigation expansion has regionally masked the historical warming of hot extremes from anthropogenic greenhouse gas emissions and other climate forcings. Using global gridded temperature observations and dedicated climate model experiments, we found that observed temperature changes as well as the probability of extreme high temperatures over intensely irrigated regions, and South Asia in particular, were reduced by a similar magnitude as the global warming signal, leading to little or no overall observed response

(Fig. 3). The consistent signal in both observations and simulations (Fig. 1) thereby increases the confidence in our results.

This study highlights the unintended beneficial impact of historical increases in irrigation on human exposure to hot extremes in several parts of the world. While the irrigation-induced cooling is mostly limited to irrigation hotspots, these are often located in densely populated areas (Supplementary Fig. 1). Thanks to the effective dampening of trends in hot extremes over irrigation hotspots, we estimate that between 0.79 and 1.29 billion people were less exposed to extreme temperatures around the year 2000 (see 'Methods'). Likewise, irrigated crops, which account for more than 40% of global yields[42], also benefited from capped temperature extremes. Yet these favourable influences only occurred because the irrigation extent more than quadrupled during the 20th century[17]. It is unsure whether this evolution will continue in the next decades: besides a possible stagnation[2,17] or even decrease[28] in the global area being irrigated, agricultural water use may potentially also become more efficient to meet sustainable development goals related to water resources availability, food security and biodiversity[33,43]. Even though changes in future irrigation extent and amounts may have important implications in densely populated irrigation hotspots, current-generation Earth system models generally ignore irrigation in climate projections[13,14,44]. These uncertainties underline the need for including transient irrigation in historical and future climate simulations, and for testing the climate response to various scenarios of future irrigation extent and irrigation efficiency.

## Methods

**Temperature and irrigation data.** We analyse temperature data from CRU TS v4.02 (hereafter referred to as CRU), which provides monthly averages of mean temperature, diurnal temperature range, daily maximum temperature and daily minimum temperature at a spatial resolution of $0.5° \times 0.5°$ between 1900 and 2017. The former two variables are gridded from station observations, while minimum and maximum temperatures are computed assuming a symmetric temperature distribution centred around the mean. We consider the average maximum temperature during the hottest month of the year (TXm), and average this metric for two 30-year periods representing the present-day (1981–2010) and the early 20th century (1901–1930). Note that the long time span of the analysis period impedes the use of satellite-based surface or air temperature datasets[45,46], which have recently been shown to be powerful tools for assessing the local biophysical effects of land cover changes[4,47]. We therefore use satellite data only for assessing present-day effects of irrigation (see Supplementary Note 2).

Irrigated area data are derived from the Historical Irrigation Dataset (HID)[17], which compiles area equipped for irrigation data based on national and sub-national statistics at a spatial resolution of $5' \times 5'$ and a temporal resolution of 10 years, increasing to 5 years after 1980. We converted area equipped for irrigation to the irrigated cell fraction and subsequently remapped the HID to the CESM grid using second-order conservative remapping[48].

**Climate simulations.** We simulate the influence of irrigation on temperature extremes using version 1.2 of the CESM, a fully coupled, state-of-the-art Earth system model. The land surface in CESM is represented by version 4.0 of the Community Land Model (CLM)[49], whose interactive irrigation module computes irrigation demand based on vegetation state and soil moisture content in a separate soil column for irrigated crops. Whenever soil moisture is limiting photosynthesis during the growing season, irrigation is activated, and the deficit between the actual and a target soil moisture content applied to the ground surface in a way that mimics extraction from nearby rivers. Although confined to the crop growing season, timing and quantities of irrigation are not prescribed, but internally computed by the irrigation parameterisation in CLM[10,49]. In terms of irrigation seasonality, this results in most irrigation hotspots receiving maximum irrigation amounts during boreal summer months, whereas South and Southeast Asia regions receive most irrigation during boreal spring.

We use CESM to generate four climate ensembles with five members each. A first control ensemble (CTL) for the period 1976–2010 (35 years including 5 years spin-up) contains all relevant climate forcings except irrigation. The second ensemble (IRR) is identical to the control experiment, except that the irrigation module is switched on. Ensembles three and four repeat this set-up, but for the period 1896–1930 (CTL_20C and IRR_20C, respectively). An extensive evaluation[10,50] of the CTL and IRR ensembles revealed that accounting for realistic irrigation in CESM leads to a small yet robust increase in model skill.

All simulations are conducted at a horizontal resolution of $0.90° \times 1.25°$, with prescribed transient greenhouse gas concentrations, sea surface temperatures and

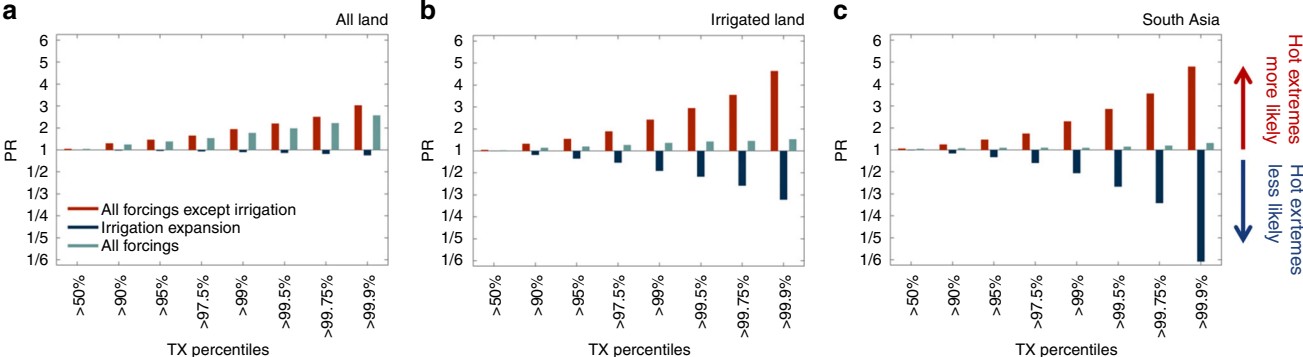

**Fig. 3 Regional masking of trends in hot extremes due to irrigation.** Median probability ratio (PR) for individual daytime temperature (TX) percentiles considering all forcings except irrigation (red), irrigation only (dark blue), and all forcings including irrigation (light blue) for all land (**a**), all irrigated land (**b**), and South Asia (**c**). Irrigated land is defined here as all pixels with >10% irrigated crop fraction, and corresponds to ∼5% of all land area. South Asia is defined as Pakistan, India, Nepal and Bangladesh, and represents ∼3% of all land. Note that the bars are non-additive because of differences in the reference ensemble.

sea ice fractions, as well as with satellite-derived vegetation phenology imposed in CLM4.0. The phenology prescription uses leaf area index, stem area index and vegetation height values derived from the Moderate Resolution Imaging Spectroradiometer (MODIS) for the period 2001–2003 and averaged to monthly climatologies at 0.05° spatial resolution[49,51,52]. By using land cover maps including irrigation extent in 1915 and 2000, respectively, our simulations capture the recorded[17] massive increase in irrigation area from 0.80 to $2.66 \times 10^6$ km$^2$ during this period. While the prescribed irrigation area represents area equipped for irrigation[17,53], CESM in practice applies actual irrigation in those areas. We note however that this does not induce a bias in irrigation quantities[10] and that this approach is common in climate model experiments including irrigation[7–9,11,12]. Moreover, climate change and irrigation expansion did not induce a notable change in the hottest month of the year (Supplementary Fig. 13). We prescribe land cover fractions other than irrigated croplands using a map for the year 2000, implying that we ignore biogeophysical effects of land cover and land management changes besides irrigation[50], but account for their biogeochemical effects through the time-varying greenhouse gas concentrations. On the first simulation day, we enforce small random perturbations on the order of 10–14 K to the initial temperature field to obtain five realisations unique in terms of natural variability but identical in terms of underlying physical processes[10].

**Window searching algorithm.** We apply a spatial window searching algorithm[20] to CRU global gridded temperature data and CESM output to reconstruct the local influence of irrigation on historical changes in monthly *TXm* and to assess the suitability of the model for determining irrigation effects on daily temperature extremes. The approach is designed to disentangle homogeneous climate forcings such as enhanced greenhouse gas concentrations from heterogeneously distributed local climate drivers including irrigation. The latest versions[20,47] of the method thereby overcome the dependence on an earlier version[19] on the binary categorisation of 'irrigated' and 'non-irrigated' grid cells, by fitting multiple linear regressions between observed temperature changes and changes in irrigation fraction within spatially moving windows.

In practical terms, the approach consists of three steps. In the first step, all eligible pixels are selected. Pixels containing a non-zero present-day irrigation fraction are selected for analysis if a window of $11 \times 11$ grid cells centred around the pixel of interest has a data coverage of at least 60%, and if at least 8% of the pixels include some irrigation. The selection criteria match those from earlier studies[20], except for the number of pixels in the search window which is larger in the present study to compensate for the higher resolution of the underlying datasets. In particular, we apply a window size of $11 \times 11$ grid cells to CESM native resolution (0.90° × 1.25°) and to the CRU output which we first regridded to the CESM grid using second-order conservative remapping[48]. As such we employ a similar window area as a previous study applying a 5 × 5 window to output from the Coupled Model intercomparison Project Phase 5 (CMIP5)[20]. While a larger search window enables to better capture the temperature contrast between irrigated cells and their surroundings, it also enhances auto-correlation[47] and spatially smooths local variations of the signal (Supplementary Note 1). Yet overall the results shown in Fig. 1 demonstrate limited sensitivity to the size of the search window compared with the change in irrigated fraction (Supplementary Fig. 14), suggesting that auto-correlation has only limited influence on our results.

The second step involves the application of the multiple linear regression technique. For each selected pixel *i* the analysis extracts an irrigation-induced temperature signal by performing a multiple linear regression on all pixels in the search window centred on *i*. The regressors predicting the total temperature change $\Delta TXm$ are the change in irrigated fraction ($\Delta f_{irr}$) and three spatial predictors which may confound the irrigation-derived signal: latitude (lat), longitude (lon) and

elevation (elev), such that:

$$\Delta TXm = \beta_1 \times \Delta f_{irr} + \beta_2 \times \text{lat.} + \beta_3 \times \text{lon.} + \beta_4 \times \text{elev.,} \quad (1)$$

with $\beta_i$ the regression coefficients for each of the four spatial predictors, and $\Delta f_{irr}$, lat, lon and elev vectors containing up to 121 values representing the conditions within the search window. The selection of the three spatial predictors next to irrigation expansion was informed by earlier tests[20] showing that inclusion of these factors succeeds in filtering out the most important natural climate gradients within the search window.

In the third step, the irrigation impact is reconstructed. The irrigation-induced temperature change in the centre pixel of the moving window $\Delta TXm_{irr}(i)$ is then obtained by multiplying the regression coefficient for irrigation $\beta_1$ from Eq. 1 by the recorded, 20th century change in irrigated fraction in that pixel:

$$\Delta TXm_{irr}(i) = \beta_1 \times \Delta f_{irr}(i). \quad (2)$$

The algorithm is designed to capture the local cooling effect of irrigation due to the enhanced surface evaporative fraction. As secondary climatic effects of irrigation such as enhanced atmospheric moisture content and cloud cover have non-local consequences (for example by modifying precipitation patterns[9–12] or altering monsoon circulation[7,8,10,11]), these are not expected to be accounted for by the algorithm. We however note that the direct effect of irrigation on latent and sensible heat flux is generally recognised as the dominant pathway through which irrigation affects near-surface climate[7–10,54–56], despite one study that attributes the irrigation effects on temperature mostly to indirect effects including changes in cloud cover and associated surface net radiation[57]. Moreover, a recent study[58] investigating direct and indirect biogeophysical effects of deforestation shows that the window searching method succeeds at reproducing factorial deforestation experiments over most deforested regions, which again suggests that ignoring indirect effects in the window searching algorithm does not strongly deteriorate the results over irrigation hotspots. In addition, this method does not correct for possible dependency between spatial predictors. For instance, irrigation is more likely to occur in lower-lying areas in the landscape due to the proximity of surface waters. Such a spatial dependence between $f_{irr}$ and elev could reduce the regression coefficient for $\Delta f_{irr}$ and thus $\Delta TXm_{irr}$, which is one of the reasons why this method is considered to be rather conservative[20]. On the other hand, it limits the false attribution of variations in temperature caused by strong natural climatic gradients within a search window (e.g. due to orography) to changes in irrigation extent. Overall, when applied to studying the historical effect of deforestation on local temperature this method was shown to give similar results than the comparison of factorial experiments[20].

Finally, we also applied our analysis to an earlier version of the temperature dataset (CRU TS v3.22). The main difference between both versions is a new interpolation algorithm, not an increase in the number of stations included in the data product. The results based on the earlier CRU version show a substantially stronger irrigation-induced cooling signal, highlighting that, next to the model simulations, also the observation-based estimates are characterised by uncertainty and thus need to be treated with care.

**PR computation.** To further analyse the climate simulations we use the PR[21] metric, which is defined as

$$PR = \frac{P_{new}}{P_{ref}}, \quad (3)$$

where $P_{ref}$ is the probability of exceeding a certain quantile in the reference ensemble—that is, 0.01 for the 99th percentile—and $P_{new}$ the probability of exceeding that quantile in the new ensemble. For instance, a PR of 2 and $P_{ref}$ of

0.01 implies that days with a maximum temperature above the 99th percentile are twice as likely in the new period (i.e. a 1-in-50 days event) compared with the reference period (i.e. a 1-in-100 days event). The effect of current irrigation on the likelihood of occurrence of extreme temperatures under present-day climate is obtained by taking CTL as the reference ensemble and IRR as the new ensemble. The effects of 'all forcings' and 'all forcings except irrigation' are computed with the IRR_20C as reference ensemble and IRR and CTL as the new ensemble, respectively (note that PRs are, consequently, non-additive). The PR metric has been referred to as risk ratio or PR in previous studies[21,22], but here PR is just a ratio of frequencies of occurrence without any reference to damage, vulnerability or exposure, which are commonly part of comprehensive risk definitions[22]. All PRs values were calculated from daily maximum temperature percentiles of the pooled ensemble members, which accounts for the possibility that extremes are more prevalent in particular years or ensemble members. We compute PR values for land pixels and irrigated land pixels, which are identified as land pixels with an irrigated fraction exceeding 10% of the total grid cell area.

**Human exposure calculation**. We compute the number of people that are less exposed to hot extremes using rural and total population density data and simulated PR. We define reduced exposure as those pixels with irrigation-induced PR < 0.5 for the 99th percentile *TX*, that is, all regions for which daytime temperatures that would occur 3–4 times a year without irrigation, now instead occur only once or twice a year (Fig. 2b). For rural and total population density we use version 3.2 of the History Database of the Global Environment (HYDE) and version 4 of the Gridded Population of the World (GPW) dataset, respectively, with values in the latter product adjusted to match United Nations country totals. Data for the year 2000 (i.e. corresponding more or less to the middle of the CTL and IRR simulation period) available at 5′ (HYDE) and 2.5′ (GPW) resolution were remapped to the CESM grid using second-order conservative remapping. We then convert population density to number of people per grid cell using CESM pixel area and land fraction values, and subsequently apply the exposure mask to compute the total number of people less exposed to hot extremes. While our upper estimate assumes that all people living within a 0.90° × 1.25° grid cell are equally exposed to the grid cell mean PR, our lower bounds neglects that the rural irrigation signature will much stronger and thus widespread[10].

## Data availability
All materials that have contributed to the reported results are available upon request, including the raw CESM model output (20 TBytes). Correspondence and requests for materials should be addressed to W.T. (wim.thiery@env.ethz.ch). The postprocessed CESM output (1.52 GBytes) is publicly available in the Figshare repository at https://doi.org/10.6084/m9.figshare.8041265. The CRU data are available at https://crudata.uea.ac.uk/cru/data/hrg/cru_ts_4.02/, the HID data at https://doi.org/10.13019/M20599, the HYDE data at http://themasites.pbl.nl/tridion/en/themasites/hyde/download/index-2.html, the GPW data at https://doi.org/10.7927/H49884ZR, and the MODIS data at https://lpdaac.usgs.gov/products/myd11c3v006/. Together with the code (see Code availability) these data sources enable reproduction of the figures presented in this study.

## Code availability
All codes used to generate the climate simulations and subsequent analyses are available through the github repository of the Department of Hydrology and Hydraulic Engineering at VUB (https://github.com/VUB-HYDR/2020_Thiery_etal_NatComm).

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

## Acknowledgements

We greatly thank Urs Beyerle and the ETH Zurich cluster team for support with the ESM simulations. We also thank the National Center for Atmospheric Research (NCAR) for maintaining CESM and making the source code publicly available. Stefan Siebert and colleagues are acknowledged for developing and providing the Historical Irrigation Dataset, while Harry Zekollari, David De Vleeschouwer, Stef Lhermitte, Jan Lenaerts, Lukas Gudmundsson and Vincent Humphrey are thanked for the helpful discussions. W.T. was supported by an ETH Zurich postdoctoral fellowship (Fel-45 15-1). The Uniscientia Foundation and the ETH Zurich Foundation are acknowledged for their support to this research. S.I.S acknowledges support from the European Research Council (ERC) through the 'DROUGHT-HEAT' project funded by the European Community's Seventh Framework Programme (grant agreement FP7-IDEAS-ERC-617518). D.M.L. is supported by the U.S. Department of Energy grants DE-FC03-97ER62402/A010 and DE-SC0012972 and U.S. Department of Agriculture grant 2015-67003-23489.

## Author contributions

W.T., E.M.F. and S.I.S. designed the study. A.J.V. performed the observational analysis under the supervision of W.T. and S.I.S. W.T. conducted the CESM simulations, performed all model analyses and wrote the paper. D.M.L., A.L.H., M.H., Q.L. and E.L.D. contributed to the data analysis. All authors commented on the paper.

## Competing interests

The authors declare no competing interests.
