## [Peer Review File · Nature Communications]

Reviewers' comments:

Reviewer #1 (Remarks to the Author):

Major Comments

* Ponding irrigation for paddy field, commonly observed in South Asia, is not just supplying water for root of paddy and avoid dryness, but to regulate the diurnal cycle of temperature smaller for better growth of paddy and to suppress the growth of weed. In that sense, the CESM simulation of irrigation process in particular for paddy field is too simple and not realistic. Water body with vegetation, like marshes, should be considered in the land surface model, if the authors really want to simulate the irrigation effect of paddy field.

* Croplands equipped with irrigation are not irrigated whole year. Farmers will irrigate only during the planting period from seeding through harvesting, even though more complex management are applied for inundated irrigation for paddy. The reviewer suspect the one of the reasons why the authors concentrated their analysis on the hottest month could be because of the coincidence between the planting period and the hot months in each region. Description on how the authors considered the period of irrigation in a year should be included.

* Figure 1: What "irrigated fraction" mean in the abscissa axis? Does it the fraction of irrigated cropland in a cell? Can it happen more than 70 % of a grid cell is occupied with irrigated cropland? I guess not, and I suspect it corresponds to the fraction of cropland equipped with irrigation equipment. If so, the authors should prepare the area of cropland in each cell, multiplied by the fraction of cropland with irrigation equipment, and divide by the total area of the cell to get the fraction of irrigated cropland area.

* Anyway, the abscissa axis of Figure 1 should not be the absolute value of the irrigated cropland but the difference of it between 1901-1930 and 1981-2010 corresponding to the axis of ordinate (temperature change). Otherwise it is unclear why higher fraction of irrigated cropland yields decrease of temperature.

* Also it may not have much value to show Figure 1 (b) since it seems most of the cells with higher ($0.5 <$) fraction of cropland with equipment for irrigation are located in South Asia and the figures are very similar in general.

Minor Comments:

* Did the authors find any shifts in the hottest month in a year between 1901-1930 and 1981-2010 in some cells on the globe?

* Line 119-120: "more water will be applied during hot periods" is ambiguous speculation. Please clarify this with evidence.

Reviewer #2 (Remarks to the Author):

Review of "Warming of hot extremes alleviated by expanding irrigation" by Thiery et al.

Thiery et al. present an evaluation of irrigation impacts on temperature extremes over the 20th century (1976-2010 compared to 1896-1930), using observations of temperatures and irrigation extent, and an ensemble of model simulations to isolate irrigation from climate change (mostly greenhouse gas) impacts. The study is split into two parts: the first part evaluates the observed increase in temperature for the hottest month of the year, conditioned on present-day irrigation extent. The second part analyses the likelihood of increased daily temperature extremes due to irrigation extent. The main result and claim of this study is that "irrigation substantially reduced human exposure to hot extremes".

The paper is very well written and presents novel findings, which contribute to the discussion on drivers of climate change and its feedbacks and are thus of high relevance. The analysis is profound and based on state-of-the-art global model simulations (CESM). There are, however, a few minor issues that could be addressed and could help to increase the reliability of the findings. I am also adding a few rather technical questions, that weren't clear to me.

Reliability of the model, its coupling strength and applied irrigation.

How realistic is the land-atmosphere coupling strength of CESM for the simulations? In other words, could the impact of dry soils -without irrigation- on daily temperature extremes be overestimated? Similarly, how realistic is the applied irrigation, not only in extent, but also in amount? Could irrigation be overestimated and the resulting effect on temperatures be overestimated? Other studies, such as Leng et al. (2015) have also shown that fluxes and states are sensitive to model parameters and input data, such as irrigation. In my opinion, some kind of significance testing or uncertainty analysis with respect to model setup and irrigation amounts could help to increase the reliability of the findings.

Novelty / Background information.

The evaluation of irrigation impacts on (daily) temperature extremes is not entirely novel, although earlier studies were mostly conducted at the regional scale (cf. e.g., Kueppers et al. 2007, Lobell et al. 2008 and Lu and Kueppers, 2015). They do, however, also analyze the impact of irrigation on daily temperature extremes, or heat indices, which may be acknowledged in the introduction.

Minor questions.

- Is the likelihood ratio computed on the ensemble members or the ensemble mean?
- Are the 3-4 hottest days of a year consecutive days, i.e. as part of a heatwave evolution, or could these extremes be artefacts of the model?
- The authors mention two feedback pathways, namely the direct feedback pathway in which irrigation increases evaporation and decreases near-surface temperature, but also the indirect feedback pathway, in which increased evaporation affects atmospheric water vapor contents, cloud formation and potentially precipitation, which may in turn affect temperatures. I assume that the direct feedback has the largest impact, but I would like to ask the authors to expand on this as it is not addressed in the manuscript.

References

Kueppers, L. M., Snyder, M. A., & Sloan, L. C. (2007). Irrigation cooling effect: Regional climate forcing by land-use change. *Geophysical Research Letters*, 34(3).

Lobell, D. B., Bonfils, C. J., Kueppers, L. M., & Snyder, M. A. (2008). Irrigation cooling effect on temperature and heat index extremes. *Geophysical Research Letters*, 35(9).

Lu, Y., & Kueppers, L. (2015). Increased heat waves with loss of irrigation in the United States. *Environmental Research Letters*, 10(6), 064010.

Reviewer #3 (Remarks to the Author):

Review for manuscript NCOMMS-18-37156-T

This study attempts to quantify the alleviating effect that the increase of irrigation has had on hot extremes in the 20th century. It further argues how this effect should be seriously incorporated into climate models as this co-benefit of irrigation partly masks the real increases in extremes, and this cooling effect will likely be lost in the future. This is pleasing to read as it is well-written and to-the-point. The subject is certainly important and deserves to be addressed. My concerns about the paper are detailed below.

First, I have some reticence about the use of CRU TS data as the suitable observations for this study. CRU data come from gridded weather station data, but these stations are unlikely to be well distributed at the right places within irrigated perimeters. Furthermore, as they must follow strict guidelines, i.e. to be measured in a standard way over a grass cover, they do not fully reflect the real temperature above irrigated areas, only that of a grassland nearby that is not under the same hydric regime. The real 'observational' data to use would seem to be land surface temperature (LST) measured from remote sensing satellites. This data is readily available at finer spatial resolution than CRU and could similarly be used in the proposed spatial window searching

algorithm. What justifies NOT using LST?

Second, I have a hard time to fully understand the rationale of using the 'unmixing' moving window approach on the CESM simulations. This multiple linear regression approach assumes that, with the exception of latitude, longitude and elevation, conditions are similar over 11 x 11 grid cells, meaning over approximately 1000 km x 1500 km! This seems to be a strong assumption. Soil types in particular would change a lot over such scales, and I would imagine this could considerably impact the cooling effect of irrigation. Other environmental factors would also likely be different across such a huge window, yet the regression is only based on XYZ pixel location and irrigation fraction. This kind of unmixing approaches would seem more appropriate over finer scales, such as those of the LST data mentioned above perhaps, but they seem more difficult to defend here (even if I know co-authors have already used it in the past, but arguably over smaller 5 x 5 windows). But ultimately, one wonders if this is indeed needed in the first place? When the CESM simulations are run, as each sub-grid component has its own soil column, can't you retain both the skin temperature of the irrigated fields and that of the non-irrigated parts of the grid? The difference between these two numbers is already the value of interest, and would not require an empirical unmixing operation on top (with its associated uncertainty). I understand this unmixing has some value to apply on already existing simulations of large ensembles of models (e.g. CMIP6) avoiding such specific calculation where sometimes it is not possible, but in this case the authors are running a single model with dedicated runs specifically for this study, and with a model that explicitly separates grids in sub-grids.

And third, in my opinion the study does not go far enough. With the authors proven capacity to run coupled climate simulations, and with the strong claims about the expected impact on future climate, one would expect to see results from future climate simulations under different scenarios of irrigation. This would provide a much stronger message as it could serve to guide policy-makers and land managers regarding what to do now about the future problem. Although it could be argued that this is heavy workload that is beyond the scope of the study, again, the authors have already shown they have the capacity to do these simulations, so it seems to be a pity to stop short from a much stronger message.

Further more punctual points are listed below:

L29: The words "but only because" seem odd. Why "only"? Is it necessary that the irrigation expanded to say this? It is just because we currently irrigate so much, irrespective of the expansion, no?

L47: Perhaps a reference to FAO would be more authoritative to state these figures

L95-96: This definition is not fully clear here in the text. Why 3-4 days? Why not say "the number of days exceeding the local 99th TX percentile"?

L102: "reverses" may be more appropriate than "offsets" in this case

L170: CRU TS is now at version 4.02 while version 3.22 is used here. Could/should the present work be updated, or could this be discarded as irrelevant?

L172: are all these monthly averages? Or daily?

L182: can you provide a reference for this 'second-order conservative remapping'?

L188-190: How realistic is this of real irrigation practices? Is irrigation in places such as South Asia, particularly in densely irrigated areas such as Punjab, not also limited by restrictions of water use from the regional government?

L200: Can you be a bit more specific about how satellite-derived vegetation phenology is imposed in CLM? Is it prescribed as a climatological mean season? But based on what period and what sensor?

L204: Given the large land cover changes that have been occurring during the 20th century, and given that biophysical effects of land cover change such as deforestation have been estimated by co-authors (and other researchers), could this assumption not be alleviated by including these biophysical effects as well?

L216: Are the spatially moving windows sliding over every pixels are they skipping around to avoid spatial autocorrelation? If the former, what is being done to remove the possible strong spatial autocorrelation?

L216: is the moving window applied to the CRU data at fine spatial resolution or at the regriding resolution of the CESM simulations?

L226: it would be welcomed to see this is supplementary material

L233: language is a bit too uncertain in: "are likely mostly not accounted for". How about being more direct: "are not expected to be accounted for by the algorithm" ?

L244-249: It would help if you give a clear example of how to interpret the LR metric, e.g.: "a LR of 4 means temperatures above the 99th percentile are 4 times more likely now than before"

L256-257: phrasing seems to suggest that "irrigated pixels" are not "land pixels"

L290: typo in author's initial: A.M.V?

Figure 1:

- change the word 'cooling' in the y-axis to 'change in temperature' to avoid possible confusion.
- How do the authors explain the considerable difference between models and observations in the more irrigated pixels?

Warming of hot extremes alleviated by expanding irrigation

April 25, 2019

Wim Thiery^{1, 2}, Auke J. Visser³, Erich M. Fischer¹, Mathias Hauser¹, Annette L. Hirsch⁴,
David M. Lawrence⁵, Quentin Lejeune⁶, Edouard L. Davin¹, Sonia I. Seneviratne¹
wim.thiery@env.ethz.ch

¹ETH Zurich, Institute for Atmospheric and Climate Science, Universitaetsstrasse 16, 8092 Zurich, Switzerland.

²Vrije Universiteit Brussel, Department of Hydrology and Hydraulic Engineering, Pleinlaan 2, 1050 Brussels, Belgium.

³Wageningen University, Meteorology and Air Quality group, Droevendaalsesteeg 3a, 6708PB, Wageningen, the Netherlands.

⁴University of New South Wales, ARC Centre of Excellence for Climate Extremes, 2052 Sydney, Australia.

⁵National Center for Atmospheric Research, Boulder, Colorado, USA.

⁶Climate Analytics, Ritterstrasse 3, 10969 Berlin, Germany.

Contents

1	Reviewer 1	3
	Reviewer 1 Comment 1	3
	Reviewer 1 Comment 2	7
	Reviewer 1 Comment 3	8
	Reviewer 1 Comment 4	12
	Reviewer 1 Comment 5	12
	Reviewer 1 Comment 6	12
	Reviewer 1 Comment 7	14
2	Reviewer 2	15
	Reviewer 2 Comment 1	15
	Reviewer 2 Comment 2	15
	Reviewer 2 Comment 3	19
	Reviewer 2 Comment 4	19
	Reviewer 2 Comment 5	19
	Reviewer 2 Comment 6	20
3	Reviewer 3	22
	Reviewer 3 Comment 1	22
	Reviewer 3 Comment 2	22

Reviewer 3 Comment 3	23
Reviewer 3 Comment 4	27
Reviewer 3 Comment 5	29
Reviewer 3 Comment 6	30
Reviewer 3 Comment 7	30
Reviewer 3 Comment 8	31
Reviewer 3 Comment 9	31
Reviewer 3 Comment 10	31
Reviewer 3 Comment 11	32
Reviewer 3 Comment 12	32
Reviewer 3 Comment 13	33
Reviewer 3 Comment 14	33
Reviewer 3 Comment 15	33
Reviewer 3 Comment 16	34
Reviewer 3 Comment 17	34
Reviewer 3 Comment 18	35
Reviewer 3 Comment 19	35
Reviewer 3 Comment 20	35
Reviewer 3 Comment 21	35
Reviewer 3 Comment 22	36
Reviewer 3 Comment 23	36

Abstract

We would like to thank all reviewers for their dedicated time reviewing the manuscript and for their useful and constructive suggestions. We carefully addressed all comments by the reviewers and the manuscript has strongly benefited from the proposed changes. We performed several additional analyses, including sensitivity experiments with CLM and land-atmosphere coupling calculations, which altogether resulted in 7 additional figures in the SI and a substantially revised manuscript and SI text. Below, we would like to clarify our changes regarding all comments, which are repeated in grey boxes.

This response letter contains numbered illustrations and references to these illustrations. To prevent confusion, the figures embedded within this response letter are called illustrations. Finally, the following convention is applied to denote modifications in the original manuscript: new text.

1 Reviewer 1

Reviewer 1 Comment 1

Ponding irrigation for paddy field, commonly observed in South Asia, is not just supplying water for root of paddy and avoid dryness, but to regulate the diurnal cycle of temperature smaller for better growth of paddy and to suppress the growth of weed. In that sense, the CESM simulation of irrigation process in particular for paddy field is too simple and not realistic. Water body with vegetation, like marshes, should be considered in the land surface model, if the authors really want to simulate the irrigation effect of paddy field.

Response

We thank the reviewer for raising this important point. Various irrigation techniques are now increasingly being implemented into land surface models, and have notably been tested in the land surface models LPJmL (Jägermeyr et al., 2015), LIS (Lawston et al., 2015) and ACME (Leng et al., 2017). Unfortunately, representation of variable irrigation techniques is currently not available in CLM. While implementing such technique is beyond the scope of the current study, we note that we currently have a research proposal under review to implement various irrigation methods in CLM5 (see also our response to Reviewer 3 - Comment 3), and will therefore hopefully be able to tackle this issue in future work together with experts on this topic. Flood irrigation code may thereby build on the natural wetland modules that were updated in the latest version of CLM (Lawrence et al., 2018). Despite this limitation and the sensitivity of the irrigation scheme to parameter settings (Sacks et al., 2009; Leng et al., 2013), the current CLM4 parameterisation with default settings was found able to realistically represent reported irrigation quantities, notably in paddy field regions of South Asia (Illustration 1 and 2; Thiery et al., 2017). Moreover, a comprehensive evaluation of the performance of the CTL and IRR simulations by Thiery et al. (2017) reveals that irrigation has a small yet overall beneficial effect on the representation of present-day near-surface climate (Illustration 3; Thiery et al., 2017).

Given the context of an imperfect irrigation scheme (see Section 4 of Thiery et al. (2017) for an extensive discussion), sensitivity experiments can help to quantify uncertainties associated with model limitations. We therefore conduct a number of sensitivity experiments with CLM4 in a land-only configuration. Note that conducting sensitivity experiments with a coupled climate model is prohibitive, since one needs to conduct ensemble simulations over a climatic period (e.g. 30 years) to get robust responses. In our case, for instance, such simulations amounted to about 3 weeks of computation time on 1500 CPUs.

To quantify the possible effect of parameter settings on modelled irrigation quantities, we tested CLM's sensitivity to the only free parameter in the irrigation module, the weighting factor in the target soil moisture calculation (see eq. 20.7 in Oleson et al. (2013)). In particular, we set it to values 0 (target soil moisture content = removal of water stress), 0.3, 0.7 (default) and 1.0 (target soil moisture content = saturation soil moisture). We ran 33-year offline CLM simulations (1972-2004) using bias-corrected reanalysis data as atmospheric forcing (Qian et al., 2006) and exclude the first three years from the analysis.

[redacted]

[redacted]

Illustration 4: Irrigation amounts for different values of the irrigation factor (see eq. 20.7 in (Oleson et al., 2013)), for (a) $\sim 1^\circ$ and (b) $\sim 2^\circ$ land-only simulations with CLM4. Values of the irrigation factor were set to 0 (target soil moisture content = removal of water stress), 0.3, 0.7 (default) and 1.0 (target soil moisture content = saturation soil moisture), respectively.

We find high sensitivity of regional irrigation quantities to the irrigation factor in most regions (Illustration 4). In South Asia (SAS), a change in the irrigation factor lead to strong biases in modelled irrigation quantities (and associated biases in climatic impacts in case this would have been run in AMIP mode). We note however that this parameter was originally set to match global observed irrigation quantities around the year 2000, which is confirmed by our IRR simulation. We also note that this sensitivity is independent of model resolution: both the $\sim 1^\circ$ and $\sim 2^\circ$ offline simulations demonstrate similar sensitivities (Illustration ref-fig:offline). We therefore conclude that irrigation quantities in most regions, and notably in the rice paddy fields of South Asia, are adequately reproduced by keeping the only free parameter in the irrigation module to its default value.

We now added a discussion of the limitations of the current version in the method section:

While conducted with a state-of-the-art Earth system model and with great care of achieving realism, our simulations remain characterized by a number of limitations. To start, we assume fixed irrigation extent (of the year 1915 and 2000, respectively) in our otherwise transient simulations as CLM4 cannot handle transient irrigation area. Furthermore, we consider only one crop type (generic C3 crops) and we do not account for variations in irrigation water sources (such as groundwater pumping or rainwater tanks (Demuzere et al., 2014; Leng et al., 2015)), and irrigation techniques (such as sprinkler, drip or flood irrigation (Jägermeyr et al., 2015; Lawston et al., 2015; Leng et al., 2017)). Instead CLM extracts required irrigation amounts from surface runoff and applies it directly to the soil surface. We thus ignore potential local water availability limitations, but note that regional patterns of applied irrigation amounts are realistic when compared to census data (Thiery et al., 2017). Moreover, a set of land-only sensitivity experiments confirms that the default irrigation parameter settings are most suited for representing irrigation quantities in South Asia (See Supplementary Information).

and we included a description of our sensitivity experiment in the supplementary informa-

tion:

To quantify the possible effect of parameter settings on modelled irrigation quantities, we tested CLM's sensitivity to the only free parameter in the irrigation module, the weighting factor in the target soil moisture calculation (see eq. 20.7 in (Oleson et al., 2013)). In particular, we set it to values 0 (target soil moisture content = removal of water stress), 0.3, 0.7 (default) and 1.0 (target soil moisture content = saturation soil moisture). We ran 33-year offline CLM simulations (1972-2004) using bias-corrected reanalysis data as atmospheric forcing (Qian et al., 2006) and exclude the first three years from the analysis.

We find high sensitivity of regional irrigation quantities to the irrigation factor in most regions (Figure 4). In South Asia (SAS), a change in the irrigation factor leads to strong biases in modelled irrigation quantities (and assumed associated biases in climatic impacts in case the simulations would have been run in coupled mode). We note however that this parameter was originally set to match global observed irrigation quantities around the year 2000, which is confirmed by evaluation of our IRR simulation (Thiery et al., 2017). We also note that this sensitivity is independent of model resolution: both the $\sim 1^\circ$ and $\sim 2^\circ$ offline simulations demonstrate similar sensitivities (Fig. 4). We therefore conclude that irrigation quantities in most regions, and notably in the rice paddy fields of South Asia, are adequately reproduced by keeping the only free parameter in the irrigation module to its default value.

Reviewer 1 Comment 2

Croplands equipped with irrigation are not irrigated whole year. Farmers will irrigate only during the planting period from seeding through harvesting, even though more complex management are applied for inundated irrigation for paddy. The reviewer suspect the one of the reasons why the authors concentrated their analysis on the hottest month could be because of the coincidence between the planting period and the hot months in each region. Description on how the authors considered the period of irrigation in a year should be included.

Response

We advance two reasons why the irrigation cooling effect is more pronounced during warm periods. First, as correctly noted by the reviewer, is the correlation between monthly temperature and irrigation quantities mediated by the growing season dependency on temperature and in many regions also by precipitation deficits (Mueller and Seneviratne, 2012). We illustrate this by plotting irrigation rates against binned daily mean temperatures for various SREX regions (Illustration 5), and included this new figure in the revised supplementary information of the manuscript. Secondly, we note that intensely irrigated regions tend to coincide with hotspots of soil-moisture temperature coupling (compare e.g. Illustration 1 to Fig. 1 in Koster et al. (2006)). In those regions soil moisture strongly constrains evapotranspiration variability and thereby atmospheric responses (transitional soil moisture regimes; Seneviratne et al., 2010). Artificially watering soils in such regions thus reduces the overall susceptibility of the local climate to temperature fluctuations.

We updated the manuscript accordingly:

Illustration 5: Irrigation rates Q_{irr} as a function of temperature in SREX regions western North America (WNA, **a**), central North America (CNA, **b**), southern Europe and Mediterranean (MED, **c**), West Asia (WAS, **d**), South Asia (SAS, **e**) and East Asia (EAS, **f**) (Seneviratne et al., 2012). Q_{irr} was binned according to daily mean temperature using a bin width of 5%; bar heights indicate the change in bin median and whiskers the change in 25th and 75th percentile of spatially averaged, daily values. The binning was performed on the individual ensemble members and bin statistics were subsequently averaged over all ensemble members. Note the varying y-axis range.

This can be understood from the nature of irrigation and land-climate dynamics: first, more water is applied during hot days (Supplementary Fig. 5) as their occurrence typically coincides with crop growing seasons and in many regions also with precipitation deficits (Mueller and Seneviratne, 2012). Second, irrigation will reduce the land-atmosphere coupling strength of a given region and thereby its sensitivity to temperature variability (Seneviratne et al., 2006) (Supplementary Fig. 7; Supplementary Information).

Reviewer 1 Comment 3

Figure 1: What "irrigated fraction" mean in the abscissa axis? Does it the fraction of irrigated cropland in a cell? Can it happen more than 70% of a grid cell is occupied with irrigated cropland? I guess not, and I suspect it corresponds to the fraction of cropland equipped with irrigation equipment. If so, the authors should prepare the area of cropland in each cell, multiplied by the fraction of cropland with irrigation equipment, and divide by the total area of the cell to get the fraction of irrigated cropland area.

Response

"Irrigated fraction" refers to the fraction of a grid cell occupied by cropland equipped with irrigation infrastructure. Indeed the Historical Irrigation Data set (HID; Siebert et al., 2005, 2015) provides "Area Equipped for Irrigation" (AEI) information which does not automatically imply that irrigation is actually taking place within those zones. In the model, however, the irrigation module is switched on for every irrigated crop tile defined by the AEI data set, implying in practice that at least some irrigation is taking place in 99% of the pixels with non-zero AEI (IRR simulation). Hence "AEI fraction" and "irrigated fraction" may be used interchangeably in case of the model simulations. For the observations this does not hold. However, until a transient data set providing actual irrigated land area becomes available, HID remains the most appropriate data source for this analysis. This is now clarified in the revised figure caption, the results and the methods.

Hereafter we summarize the changes related to figure 1:

- We clarified term irrigated fraction in the figure caption and methods section.
- We further modified the figure caption to enhance readability.
- We now harmonized the calculation of Δf_{irr} in CESM and CRU. This change led to decreased regression coefficients for the CESM results and thus resulted in a small decrease in simulated irrigation-induced temperature change. We note however that overall this change had very limited effect on the overall results.
- We changed the abscissa axis of Fig. 1 from f_{irr} to Δf_{irr} .
- We discovered a bug in the netcdf file used as input for the production of the figure. The bug is related to the CDO version used to postprocess the raw CESM output, and was reported by us to the CDO team at MPI when we discovered it in the context of another study (<https://code.mpimet.mpg.de/issues/8465>, login required). Resolving this bug (i.e. changing the CDO version) substantially increased the correspondence between model and observations in panels a and b of the figure, while having little influence on the results of the window-searching algorithm shown in panels c and d.
- We adapted panels e and f of Fig. 1, these now show the number of pixels per Δf_{irr} -bin meeting the selection criteria for computing the irrigation-induced cooling (i.e. the bin sizes of panels c and d). In the original version they were showing the bin sizes of panels a and b.
- We now made the code to generate this and all other figures open-access through the github page of the Department of Hydrology and Hydraulic Engineering at VUB (<https://github.com/VUB-HYDR>).
- We updated the observational analysis using the latest version of the CRU data set (v4.02 instead of v3.22). The results are qualitatively consistent, but where the irrigation-induced cooling in v3.22 closely matches the values obtained from CESM (Illustration 8c-d), the results from v4.02 yield a smaller cooling effect (Illustration 7c-d). Overall, this highlights the importance of observational uncertainty, especially if we assume that the main difference between both CRU versions lies in the postprocessing of the raw data (see also Supplementary Information). This strengthens our initial decision

Illustration 6: Figure 1 at time of initial submission.

to combine both observational data and model simulations. In the revised manuscript, we show the results with v4.02 in the main paper and the results with v3.22 in the SI.

While these modification increase the reliability of the analysis and robustness of Fig. 1, we note that the main messages arising from the figure did not change (compared Illustration 6 to illustration 7).

Caption Fig. 1: Observed warming rates affected by irrigation. Boxplots of total TXm change (ΔTXm , upper panels) and irrigation-induced temperature change (ΔTXm_{irr} , middle panels) for global land (left panels) and South Asia (right panels) between 1901-1930 and 1981-2010, binned by the change in grid cell fraction equipped for irrigation (hereafter referred to as irrigated fraction f_{irr}) between both periods. Cell counts per bin for ΔTXm_{irr} are indicated in the lower panels. Blue bars represent results from the CRU observational data, orange bars show results from the CESM global climate simulations. Note that most grid cells with $f_{irr} > 0.5$ for the global land are situated in south Asia.

Results: While constituting only a small portion of the global land area (0.5%), pixels with an increase in irrigated grid cell fraction above 35% generally experienced a cooling trend. [...] Results of applying the reconstruction method indicate that irrigation expansion has a cooling influence on TXm which increases with greater irrigation extent (Fig. 1c,d), and reaches almost 1 K for the most intensely irrigated regions when using CRU version 3.22 (Supplementary Fig. 2).

Methods: While the prescribed irrigation area represents area equipped for irriga-

Illustration 7: Figure 1 at time of revised submission.

Illustration 8: Same as Figure 1 at time of revised submission but with the original CRU version 3.22 (Supplementary Fig. 2 in the revised manuscript).

tion (Siebert et al., 2005, 2015), CESM in practice applies actual irrigation in those areas. We note however that this does not induce a bias in irrigation quantities (Thiery et al., 2017) and that this approach is common in climate model experiments including irrigation (Puma and Cook, 2010; Guimberteau et al., 2012; Cook et al., 2015; de Vrese et al., 2016; Keune et al., 2018).

At the model's native horizontal resolution of $0.90^\circ \times 1.25^\circ$, 12 pixels have a present-day irrigated grid cell fraction larger than 70%. However, by changing the abscissa axis from present-day area to area change (as requested by the reviewer), the axis range changed in the revised figure.

Reviewer 1 Comment 4

Anyway, the abscissa axis of Figure 1 should not be the absolute value of the irrigated cropland but the difference of it between 1901-1930 and 1981-2010 corresponding to the axis of ordinate (temperature change). Otherwise it is unclear why higher fraction of irrigated cropland yields decrease of temperature.

Response

We updated Figure 1 and the figure caption as requested by the reviewer. See our response to Reviewer 1 Comment 3 for all updates to Fig. 1.

Reviewer 1 Comment 5

Also it may not have much value to show Figure 1 (b) since it seems most of the cells with higher ($0.5 <$) fraction of cropland with equipment for irrigation are located in South Asia and the figures are very similar in general.

Response While we confirm that most of the cells with high fraction of AEI ($f_{irr} > 0.5$) are situated in South Asia, we believe it is nonetheless useful to maintain both panels. This is because in Figure 3 we again compare global land (panel a) to south Asia (panel c), now showing the pronounced difference between both regions in terms of spatial median LR. We have now added a caveat in the caption of Figure 1:

Note that most grid cells with $f_{irr} > 0.5$ for the global land are situated in south Asia.

Reviewer 1 Comment 6

Did the authors find any shifts in the hottest month in a year between 1901-1930 and 1981-2010 in some cells on the globe?

Response

As requested by the reviewer, we now calculated the change in hottest month in a year between both simulation periods from the *IRR* and *20C1RR* ensembles. The results indicate that there are only very few regions experiencing a shift in the hottest month, and that for those regions experiencing a shift, this is typically only by one month (Illustration 9). We included this result as a supplementary figure and now refer to it from the manuscript.

Illustration 9: Climate change and irrigation expansion have limited influence on the timing of the hottest month of the year. (a), hottest month of the year during the reference period according to the IRR_20C ensemble. (b), change in hottest month of the year from the reference period (IRR_20C ensemble) to the present-day period (IRR ensemble).

Moreover, climate change and irrigation expansion did not induce a notable change in the hottest month of the year (Supplementary Fig. 8).

Reviewer 1 Comment 7

Line 119-120: "more water will be applied during hot periods" is ambiguous speculation. Please clarify this with evidence.

Response

We now included a new figure (Illustration 5) in the Supplementary Information which clearly shows that more irrigation is applied during warm days. We also rephrased this paragraph and refer to Reviewer 1 - comment 2 for a detailed discussion of this point.

2 Reviewer 2

Reviewer 2 Comment 1

Thierry et al. present an evaluation of irrigation impacts on temperature extremes over the 20th century (1976-2010 compared to 1896-1930), using observations of temperatures and irrigation extent, and an ensemble of model simulations to isolate irrigation from climate change (mostly greenhouse gas) impacts. The study is split into two parts: the first part evaluates the observed increase in temperature for the hottest month of the year, conditioned on present-day irrigation extent. The second part analyses the likelihood of increased daily temperature extremes due to irrigation extent. The main result and claim of this study is that “irrigation substantially reduced human exposure to hot extremes”. The paper is very well written and presents novel findings, which contribute to the discussion on drivers of climate change and its feedbacks and are thus of high relevance. The analysis is profound and based on state-of-the-art global model simulations (CESM). There are, however, a few minor issues that could be addressed and could help to increase the reliability of the findings. I am also adding a few rather technical questions, that weren't clear to me.

Response

We thank Reviewer 2 for their overall support of our study. Below we address the issues that were raised for improvement of the manuscript.

Reviewer 2 Comment 2

Reliability of the model, its coupling strength and applied irrigation. How realistic is the land-atmosphere coupling strength of CESM for the simulations? In other words, could the impact of dry soils -without irrigation- on daily temperature extremes be overestimated? Similarly, how realistic is the applied irrigation, not only in extent, but also in amount? Could irrigation be overestimated and the resulting effect on temperatures be overestimated? Other studies, such as Leng et al. (2015) have also shown that fluxes and states are sensitive to model parameters and input data, such as irrigation. In my opinion, some kind of significance testing or uncertainty analysis with respect to model setup and irrigation amounts could help to increase the reliability of the findings.

Response

We have now added a new figure and section in the Supplementary Information describing the effects of irrigation on land-atmosphere coupling strength. From this analysis we conclude that it is unlikely that CESM overestimates the land-atmosphere coupling in irrigation hotspots, and can therefore be considered an appropriate tool to investigate irrigation effects on climate:

Irrigation influence on land-atmosphere coupling strength. To quantify the simulated effect of irrigation on land-atmosphere coupling strength, we apply the two-legged coupling framework (Dirmeyer, 2011; Dirmeyer et al., 2014) to

our CESM simulations. In particular, we use daily model output for three variables: 2-meter air temperature (T), sensible heat flux (SHF) and soil moisture (liquid+solid) in the top 10 cm of the soil (SM). Using these variables we compute three metrics: (i) the atmospheric leg of the coupling, defined here as the product of the standard deviation of T and the correlation between T and SHF ($\sigma_T \text{ corr}_{SHF,T}$), (ii) the terrestrial leg of the coupling, defined here as the product of the standard deviation of SHF and the correlation between SHF and SM ($\sigma_{SHF} \text{ corr}_{SM,SHF}$), and (iii) full land-atmosphere coupling, defined here as the product of the standard deviation of T and the correlation between SM and T ($\sigma_T \text{ corr}_{SM,T}$). All metrics were computed on the native CESM grid, at daily resolution for each individual year, and subsequently averaged across all 30 analysis years and ensemble members. Since most irrigation areas are located in the northern hemisphere, we focus our analysis on northern hemisphere and the months MAMJJA (Fig. 10), as such including the hottest months in most of the study area (Fig. 9a). While there are some differences between the two-legged coupling metric and the standard Global Land-Atmosphere Coupling Experiment (GLACE) metric Ω_T (Seneviratne et al., 2013; Guo et al., 2006), we note that generally coupling hotspots are similar between the various diagnostics (Lorenz et al., 2015).

The sign of each metric denotes regions where the two considered variables are correlated or anti-correlated. Hence the sign reversal between the atmospheric and terrestrial leg of the coupling highlights the general relation between both variables (Atmospheric leg: higher SHF inducing higher T; Terrestrial leg: higher SM inducing lower SHF), whereas the magnitude is indicative of the strength of the coupling, with values closer to zero indicating weaker coupling. However, overall one should be careful with the interpretation of the results of this analysis at the local to regional scale, as well as with the attribution of irrigation effects outside the main irrigation centers, considering the importance of natural variability in this context.

Over South Asia, both the atmospheric and terrestrial leg of the coupling are weakening due to irrigation activity (that is, in both cases the coupling in IRR is closer to zero compared to CTL, Fig. 10a-b,c-d), leading to an overall decrease in full land-atmosphere coupling (Fig. 10c,f). This is likely because irrigation leads to a decrease in the variability of SHF. Regarding the atmospheric leg, the decrease is particularly pronounced over South Asia, suggesting that irrigation additionally dampens temperature variability, consistent with earlier findings (Thiery et al., 2017). Furthermore, timing of irrigation during periods of vegetation water scarcity reduces temporal SM variability and associated terrestrial leg as well as full land-atmosphere coupling.

Finally, irrigation effects on climate as simulated by CESM may be erroneous in case the impact of dry soils – without irrigation – on daily temperature extremes would be overestimated in the model. In this context, we note that CESM does not highlight South Asia as a land-atmosphere coupling hotspot (Fig. 10c), in contrast to several other models participating in the GLACE studies (Guo et al., 2006). Therefore, it is unlikely that CESM overestimates the impact of dry soils on climate compared to other Earth system models.

Illustration 10: Irrigation influence on land-atmosphere coupling strength reveals reduced temperature dependence on sensible heating and soil moisture. Results from applying the two-legged coupling framework (Dirmeyer et al., 2014) to the CESM simulations. The atmospheric leg, terrestrial leg and full land-atmosphere coupling are shown for (a-c) the CTL ensemble, (d-f) the IRR ensemble, and (g-i) the difference between both ensembles. σ denotes the standard deviation, corr the Pearson correlation coefficient, T the 2-meter air temperature, SHF the sensible heat flux, and SM the soil liquid water and ice in the top 10 cm of the soil. Note the difference in color scale in the second column compared to the first and last column. See Supplementary Information for more details on the methods and interpretation of the results.

Regarding irrigation amounts, we note that, despite the limitations of the CLM4 irrigation parameterisation, the model was previously found to realistically represent reported irrigation quantities, notably over South Asia (Illustration 1 and 2; Thiery et al., 2017). We now added a discussion of the limitations and evaluation of the current irrigation parameterisation in the method section:

While conducted with a state-of-the-art Earth system model and with great care of achieving realism, our simulations remain characterized by a number of limitations. To start, we assume fixed irrigation extent (of the year 1915 and 2000, respectively) in our otherwise transient simulations as CLM4 cannot handle transient irrigation area. Furthermore, we consider only one crop type (generic C3 crops) and we do not account for variations in irrigation water sources (such as groundwater pumping or rainwater tanks (Demuzere et al., 2014; Leng et al., 2015)), and irrigation techniques (such as sprinkler, drip or flood irrigation (Jägermeyr et al., 2015; Lawston et al., 2015; Leng et al., 2017)). Instead CLM extracts required irrigation amounts from surface runoff and applies it directly to the soil surface. We thus ignore potential local water availability limitations, but note that regional patterns of applied irrigation amounts are realistic when compared to census data (Thiery et al., 2017). Moreover, a set of land-only sensitivity experiments confirms that the default irrigation parameter settings are most suited for representing irrigation quantities in South Asia (See Supplementary Information).

Finally, regarding the uncertainty analysis: we now included results from a sensitivity analysis in the Supplementary Information of the paper, (i) confirming the results from Leng et al. (2015) that the irrigation parameterisation is sensitive to parameters settings, and (ii) highlighting that the default value of the weighting factor used in the target soil moisture calculation is most appropriate regarding the simulation of realistic irrigation quantities. We refer to our response to Reviewer 1 - comment 1 for an extensive description of the rationale and results of the sensitivity analysis. We now also included a description of the set-up and results of our sensitivity experiment in the supplementary information:

To quantify the possible effect of parameter settings on modelled irrigation quantities, we tested CLM's sensitivity to the only free parameter in the irrigation module, the weighting factor in the target soil moisture calculation (see eq. 20.7 in Oleson et al. (2013)). In particular, we set it to values 0 (target soil moisture content = removal of water stress), 0.3, 0.7 (default) and 1.0 (target soil moisture content = saturation soil moisture). We ran 33-year offline CLM simulations (1972–2004) using bias-corrected reanalysis data as atmospheric forcing (Qian et al., 2006) and exclude the first three years from the analysis.

We find high sensitivity of regional irrigation quantities to the irrigation factor in most regions (Figure 4). In South Asia (SAS), a change in the irrigation factor leads to strong biases in modelled irrigation quantities (and assumed associated biases in climatic impacts in case the simulations would have been run in coupled mode). We note however that this parameter was originally set to match global observed irrigation quantities around the year 2000, which is confirmed by evaluation of our IRR simulation (Thiery et al., 2017). We also note that this sensitivity is independent of model resolution: both the $\sim 1^\circ$ and $\sim 2^\circ$ offline simulations

demonstrate similar sensitivities (Fig. 4). We therefore conclude that irrigation quantities in most regions, and notably in the rice paddy fields of South Asia, are adequately reproduced by keeping the only free parameter in the irrigation module to its default value.

Reviewer 2 Comment 3

Novelty / Background information. The evaluation of irrigation impacts on (daily) temperature extremes is not entirely novel, although earlier studies were mostly conducted at the regional scale (cf. e.g., Kueppers et al. 2007, Lobell et al. 2008 and Lu and Kueppers, 2015). They do, however, also analyze the impact of irrigation on daily temperature extremes, or heat indices, which may be acknowledged in the introduction.

Response

The reviewer is correct that earlier studies have demonstrated the cooling potential of irrigation activity on daily temperature extremes. While we are aware of these studies and cite them in our previous publication on the subject (Thiery et al., 2017), we initially had to limit the number of references to 30. In the revised manuscript we now include an additional number of relevant references:

While the influence of irrigation on annual mean temperatures remains limited, emerging evidence reveals a large impact of irrigation on temperature extremes, with a particularly strong cooling during the hottest days of the year (Bonfils and Lobell, 2007; Kueppers et al., 2007; Lobell et al., 2008; Kueppers and Snyder, 2012; Lu and Kueppers, 2015; Mueller et al., 2016; Hirsch et al., 2017; Thiery et al., 2017).

Reviewer 2 Comment 4

Is the likelihood ratio computed on the ensemble members or the ensemble mean?

Response

The LR is computed on the pooled ensemble members, that is, on the total of 5x30x365 days per time slice. We believe this is the right approach: since all ensemble members experience the same external forcing, each simulation year represents an equally plausible realisation of that climate. Moreover, in case we would have used the ensemble mean extremes would be averaged out. This is now clarified in the methods:

All *LRs* values were calculated from daily maximum temperature percentiles of the pooled ensemble members, which accounts for the possibility that extremes are more prevalent in particular years or ensemble members.

Reviewer 2 Comment 5

Are the 3-4 hottest days of a year consecutive days, i.e. as part of a heatwave evolution, or could these extremes be artefacts of the model?

Response

Application of the method does not imply that days with temperature above the 99th percentile are consecutive. Moreover, as we compute the percentiles on the entire concatenated ensemble, extremes may cluster in particular years or ensemble members (still we think is the right approach, see our response to previous comment). The statement referred to by the reviewer is included to make our approach comprehensible for experts from other fields which also consult this journal.

We are currently working on a follow-up analysis looking particularly at land management effects during heatwaves. As part of this ongoing analysis, we found that, interestingly, only about 30% of the time TXx coincides with a heatwave, with a stronger co-occurrence over high-latitude areas and less agreement for tropical regions.

Reviewer 2 Comment 6

The authors mention two feedback pathways, namely the direct feedback pathway in which irrigation increases evaporation and decreases near-surface temperature, but also the indirect feedback pathway, in which increased evaporation affects atmospheric water vapor contents, cloud formation and potentially precipitation, which may in turn affect temperatures. I assume that the direct feedback has the largest impact, but I would like to ask the authors to expand on this as it is not addressed in the manuscript.

Response

We thank the reviewer for raising this important point. We investigated the relative importance of various feedback pathways in our previous study on this subject (Thiery et al., 2017). This was achieved by applying the surface energy balance decomposition method (Luyssaert et al., 2014; Akkermans et al., 2014; Thiery et al., 2015; Duveiller et al., 2018) to the present-day CESM simulations. As an illustration, we here show the results for the SREX regions Mediterranean and South Asia (Illustration 11). Over irrigated lands in South Asia, the Mediterranean and other irrigation regions, the net temperature signal is dominated by the change in turbulent flux partitioning of the net surface radiation. A strong increase in LHF contributes to a lower surface temperature in irrigated pixels and represents the largest individual contribution to the surface temperature signal. This cooling effect is partly offset by the warming influence of Δ SHF (a reduced upward SHF leaves more energy available at the surface). Other effects have only a minor influence on surface temperatures. Wet soils are darker and therefore reflect less incoming radiation, resulting in a direct positive contribution to surface temperature. The remaining contributions to the total surface temperature change are generally negative and originate from decreasing incoming shortwave and long-wave radiation caused by changing cloud cover and atmospheric temperature.

Our results generally agree with earlier analyses on this subject (e.g., Lobell et al., 2009; Puma and Cook, 2010; Guimberteau et al., 2012; Cook et al., 2015; Kueppers and Snyder, 2012; Lu et al., 2015), but contrast with Sacks et al. (2009) who attribute the irrigation effects mostly to changes in cloud cover and associated surface net radiation. Moreover, a recent study which splits up direct and indirect biogeophysical effects of changes in tree cover (Winckler et al., 2019), shows that the window searching method succeeds at reproducing the findings

[redacted]

of the factorial experiments on the temperature effect of these changes in tree cover over most deforested regions. This is an indication that not having the indirect pathway in the window searching algorithm does not strongly affect the results over the hotspots we have identified in the case of irrigation. In the revised manuscript we now included a discussion on this topic:

The algorithm is designed to capture the local cooling effect of irrigation due to the enhanced surface evaporative fraction. As secondary climatic effects of irrigation such as enhanced atmospheric moisture content and cloud cover have non-local consequences (for example by modifying precipitation patterns (Cook et al., 2015; de Vrese et al., 2016; Thiery et al., 2017; Keune et al., 2018) or altering monsoon circulation (Puma and Cook, 2010; Guimberteau et al., 2012; de Vrese et al., 2016; Thiery et al., 2017)), these are likely mostly not accounted for by the algorithm. We however note that the direct effect of irrigation on latent and sensible heat flux is generally recognised as the dominant pathway through which irrigation affects near-surface climate (Lobell et al., 2009; Puma and Cook, 2010; Guimberteau et al., 2012; Cook et al., 2015; Kueppers and Snyder, 2012; Lu et al., 2015; Thiery et al., 2017), despite one study that attributes the irrigation effects on temperature mostly to indirect effects including changes in cloud cover and associated surface net radiation (Sacks et al., 2009). Moreover, a recent study (Winckler et al., 2019) investigating direct and indirect biogeophysical effects of deforestation shows that the window searching method succeeds at reproducing factorial deforestation experiments over most deforested regions, which again suggests that ignoring indirect effects in the window searching algorithm does not strongly deteriorate the results over irrigation hotspots.

3 Reviewer 3

Reviewer 3 Comment 1

This study attempts to quantify the alleviating effect that the increase of irrigation has had on hot extremes in the 20th century. It further argues how this effect should be seriously incorporated into climate models as this co-benefit of irrigation partly masks the real increases in extremes, and this cooling effect will likely be lost in the future. This is pleasing to read as it is well-written and to-the-point. The subject is certainly important and deserves to be addressed. My concerns about the paper are detailed below.

Response

We thank the Reviewer for the positive comments about the text, structure and importance of the study. Below, we address every comment carefully and explain the corresponding changes in the manuscript.

Reviewer 3 Comment 2

First, I have some reticence about the use of CRU TS data as the suitable observations for this study. CRU data come from gridded weather station data, but these stations are unlikely to be well distributed at the right places within irrigated perimeters. Furthermore, as they must follow strict guidelines, i.e. to be measured in a standard way over a grass cover, they do not fully reflect the real temperature above irrigated areas, only that of a grassland nearby that is not under the same hydric regime. The real 'observational' data to use would seem to be land surface temperature (LST) measured from remote sensing satellites. This data is readily available at finer spatial resolution than CRU and could similarly be used in the proposed spatial window searching algorithm. What justifies NOT using LST?

Response

We fully agree with reviewer 3 that the spatial impact of irrigation on temperature can indeed be better assessed using the spatially explicit, remotely sensed LST data. In fact, the first studies assessing the biogeophysical impacts of land cover changes using LST and other remote sensing data are now becoming available (Bright et al., 2017; Duveiller et al., 2018). We believe there is great potential for extending such analyses from land cover change to land management change (e.g. irrigation, fertilizer use, forest management), especially since the latter changes have been highlighted as having an imprint on surface energy balance of about equal magnitude as land cover changes (Luyssaert et al., 2014).

However, the main research question in this study is not related to the present-day effect of irrigation (that is, its spatial imprint), but to how this effect *evolved* over time (that is, its spatio-temporal imprint). Indeed, this is the only way in which the irrigation-induced signal can be compared against the greenhouse gas signal (Mueller et al., 2016; Lejeune et al., 2018). In this respect, we are limited to using data sets available over the course of the entire 20th century, thereby ruling out remote sensing sources. For this study, we therefore rely on

gridded surface temperature products such as CRU.

We now included a justification for using gridded station data instead of LST data in the manuscript:

Note that that the long time span of the analysis period impedes the use of satellite-based surface or air temperature data sets (Wan, 2014; Hooker et al., 2018), which have recently been shown to be powerful tools for assessing the local biophysical effects of land cover changes (Bright et al., 2017; Duveiller et al., 2018).

Reviewer 3 Comment 3

Second, I have a hard time to fully understand the rational of using the ‘unmixing’ moving window approach on the CESM simulations. This multiple linear regression approach assumes that, with the exception of latitude, longitude and elevation, conditions are similar over 11 x 11 grid cells, meaning over approximately 1000 km x 1500 km! This seems to be a strong assumption. Soil types in particular would change a lot over such scales, and I would imagine this could considerably impact the cooling effect of irrigation. Other environmental factors would also likely be different across such as huge window, yet the regression is only based on XYZ pixel location and irrigation fraction. This kind of unmixing approaches would seem more appropriate over finer scales, such as those of the LST data mentioned above perhaps, but they seem more difficult to defend here (even if I know co-authors have already used it in the past, but arguably over smaller 5 x 5 windows). But ultimately, one wonders if this is indeed needed in the first place? When the CESM simulations are run, as each sub-grid component has its own soil column, can’t you retain both the skin temperature of the irrigated fields and that of the non-irrigated parts of the grid? The difference between these two numbers is already the value of interest, and would not require an empirical unmixing operation on top (with its associated uncertainty). I understand this unmixing has some value to apply on already existing simulations of large ensembles of models (e.g. CMIP6) avoiding such specific calculation where sometimes it is not possible, but in this case the authors are running a single model with dedicated runs specifically for this study, and with a model that explicitly separates grids in sub-grids.

Response

We thank the reviewer for raising this important point. Hereafter we structure our response in several bullet points:

- To start, we have actually chosen our window size such that its total window area is comparable to the study of Lejeune et al. (2018). Lejeune et al. (2018) employed a window size of 5 x 5, but applied it to the native CMIP5 simulation output which typically has coarser resolution than the 0.90° x 1.25° grid on which we performed our analysis. We use 11 x 11 pixels at 0.90° x 1.25° resolution, whereas Lejeune et al. (2018) employed 5 x 5 pixels windows on models with resolution between 0.90° and 2.80° for the latitude, and between 1.250° and 3.750° for the longitude. We now phrased this correctly and more explicitly in the method section.

- Lejeune et al. (2018), who improved the window searching algorithm initially developed by Kumar et al. (2013), has demonstrated the validity of the approach to reconstruct regional mean historical changes in surface temperature (including extremes) due to changes in tree cover in Earth System Models by comparing the regression-based window searching method against factorial experiments (Supplementary Information of Lejeune et al. (2018)). This analysis included Earth System models with resolutions much coarser than our simulations (see previous bullet point), highlighting that the approach is valid for window sizes well above the size we used here (even if the window included less grid cells).
- In addition, we have now analysed the sensitivity of the irrigation-induced temperature change to the search window size for CESM and CRU TS v3.22 and v4.02. In Illustrations 12 and 13 we compare the sensitivity of the irrigation-induced cooling to both the change in irrigated fraction and search window size. While the results demonstrate some sensitivity to the search window size, this sensitivity is substantially smaller than the sensitivity to the change in irrigated fraction, confirming the overall validity of our approach. Interestingly, (i) CESM has less sensitivity compared to CRU, and (ii) results differ between CRU versions, highlighting the influence of the postprocessing the raw observations in the CRU data set (i.e. the main difference between both CRU versions). Overall we believe to have selected the right window size considering the coarse resolution of the underlying products and the need to have a large enough window size to derive robust coefficient estimates. We have now included both figures and a discussion on this topic in the Supplementary information.
- To further test the validity of our approach, we also tested whether deforestation may confound our results by including tree fraction cover change as predictor in our model. However, since this did not affect our results (which makes sense given that deforestation typically occurred regions outside the irrigation hotspots), we decided to omit the variable from the set of predictors.
- Another point to realize here is that we are using a methodology that can be applied to observational data where the irrigation effect is implicitly included. While we can (and do) isolate the irrigation-induced cooling from our simulation set-up, the reason for applying the window searching method to both model output and observational data is to demonstrate that CESM is an appropriate tool to estimate the ‘irrigation effect’. In short, we apply the window searching method to the model output to obtain a consistency check. We now clarify this in the ‘Window Searching Algorithm’ method section.
- Finally, CESM indeed applies a subgrid-scale approach and we have extensively analysed that output in our previous study (see e.g. illustration 14). However, while SEB terms and surface temperatures are computed at the subgrid-scale; near-surface air temperatures are only computed at the grid-scale. Since daily maximum near-surface air temperature is our variable of interest, the subgrid scale analyses offer no solution to our problem (unless LSTs can be reliably converted to T2M, as for instance done by Hooker et al. (2018)). We however note that such an approach has not yet been tested on ESM output).

We made the following changes in the manuscript:

Illustration 12: Sensitivity to changes in irrigated fraction dominates over sensitivity to search window size. Irrigation-induced daytime temperature change during the hottest month of the year across a range of sensitivity experiments with window size varying from 5×5 to 15×15 (y-axis). Results are binned according to the change in irrigated fraction (x-axis) and bin medians are shown for (a) CESM and (b) CRUv4.02.

Illustration 13: Sensitivity to changes in irrigated fraction dominates over sensitivity to search window size. Same as illustration 12, but now using CRUv3.22.

[redacted]

We converted area equipped for irrigation to the irrigated cell fraction and subsequently remapped the HID to the CESM grid using second-order conservative remapping (Jones, 1999).

The selection criteria match those from earlier studies (Lejeune et al., 2018), except for the number of pixels in the search window which is larger in the present study to compensate for the higher resolution of the underlying datasets. In particular, we apply a window size of 11×11 grid cells to CESM native resolution ($0.90^\circ \times 1.25^\circ$) and to the CRU output which we first regrid to the CESM grid using second-order conservative remapping (Jones, 1999). As such we employ a similar window area as a previous study applying a 5×5 window to output from the Coupled Model intercomparison Project Phase 5 (CMIP5) Lejeune et al. (2018).

We apply a spatial window searching algorithm (Lejeune et al., 2018) to CRU global gridded temperature data and CESM output to reconstruct the local influence of irrigation on historical changes in monthly TXm and to assess the suitability of the model for determining irrigation effects on daily temperature extremes. [...] The latest versions (Lejeune et al., 2018; Duveiller et al., 2018) of the method thereby overcome the dependence of an earlier version Kumar et al. (2013) on the binary categorisation of 'irrigated' and 'non-irrigated' grid cells, by fitting multiple linear regressions between observed temperature changes and changes in irrigation fraction within spatially moving windows.

And we now added a paragraph to the Supplementary Information:

Secondly, we have analysed the sensitivity of the irrigation-induced temperature change to the search window size for CESM and CRUv3.22 and v4.02. While the obtained irrigation-induced cooling demonstrates some sensitivity to the search window size, this sensitivity is substantially smaller than the sensitivity to the change in irrigated fraction (Figs. 12 and 13), confirming the overall validity of our approach. Interestingly, (i) CESM has less sensitivity compared to CRU, and (ii) results differ between CRU versions, highlighting the influence of the post-processing the raw observations in the CRU data set. Overall we believe to have selected the right window size considering the coarse resolution of the underlying products and the need to have a large enough window size to derive robust coefficient estimates.

Reviewer 3 Comment 4

And third, in my opinion the study does not go far enough. With the authors proven capacity to run coupled climate simulations, and with the strong claims about the expected impact on future climate, one would expect to see results from future climate simulations under different scenarios of irrigation. This would provide a much stronger message as it could serve to guide policy-makers and land managers regarding what to do now about the future problem. Although it could be argued that this is heavy workload that is beyond the scope of the study, again, the authors have already shown they have the capacity to do these simulations, so it seems to be a pity to stop short from a much stronger message.

Response

We thank the reviewer for their suggestion to consider projections to estimate the mitigation potential of future irrigation activity on heat extremes. In two recent studies led by co-authors (Hirsch et al. (2017) and Hauser et al. (2019)), we demonstrate the potential role of irrigation in altering the projections of heat extremes. In contrast to those two idealised studies, the aim of this study to represent irrigation as realistic as possible in our historical climate simulations: only in such setup we can compare the impact of historical irrigation activity to the effects of climate change. There are several limitations involving the adequate sampling of uncertainty to expand the present manuscript to include projections, in particular if one wishes to move beyond sensitivity experiments. These limitations include:

1. Climate model uncertainty
2. Natural variability uncertainty
3. Emission scenario uncertainty
4. Irrigation extent scenario uncertainty
5. Irrigation efficiency scenario uncertainty

Regarding climate model uncertainty, we would require the participation of other climate model groups to augment our existing experiment. The Land Use Model Intercomparison Project (LUMIP; Lawrence et al., 2016) aims to provide a multi-model estimation of the impact of (aggregated) future land cover and land management changes. In case the modelling teams conduct ensemble simulations (which is planned for by several groups), these simulations will also account for natural variability uncertainty. This is a considerable undertaking with first results expected in late 2019.

Regarding emission scenario uncertainty, prior research by Hirsch et al. (2018) using four Earth System Models has demonstrated that the characterization of land surface conditions (both in terms of vegetation and land management) becomes important for low emissions scenarios as land cover and land use changes impose a larger relative forcing on the surface climate. Projections of future irrigation impacts should therefore sample different emission scenarios (i.e. low, medium and high) in addition to different land use scenarios linked to conventional mitigation activities (i.e. reforestation, agricultural intensification). This endeavour is not a trivial undertaking and will be pursued in future research. In particular, the lead author just acquired funding through the JPI Climate scheme to assess the future role of irrigation while accounting for model, natural variability and emission scenario uncertainty (MPI-ESM: Julia Pongratz, EC-EARTH: Dim Coumou; CESM: Wim Thiery).

Fourth, there are different pathways through which irrigation extent may evolve towards the future. This is illustrated by the output from several Integrated Assessment Models (IAMs) that produced output for CMIP6, showing that irrigation area may double or even decrease (illustration 15). However, the version of CESM we use in this study cannot incorporate transient irrigation area changes. To run transient future projection with and without irrigation, one would need to update the CESM version and thus also re-compute all past ensembles.

Finally, sampling the uncertainty in the irrigation efficiency via different technologies (e.g. flood, drip, sprinkler) would require further source code development of the climate model.

Illustration 15: Global time series of cropland area (a), and irrigated cropland area (b) for future simulations. Lines indicate SSP-RCP scenarios chosen for ScenarioMIP, with colored lines representing scenarios with specific LUMIP experiments. Data is provided by the IAM community. Adapted from Lawrence et al. (2016) under the CC BY 3.0 licence (<https://creativecommons.org/licenses/by/3.0/>)

We currently have a research proposal under review that will address this topic.

Overall, we believe that a proper assessment of future impacts of irrigation (beyond mere sensitivity experiments) requires years of research, which we plan to conduct with the highest priority in one or two new research projects. We therefore also think that assessing the future potential of irrigation in a 'quick and dirty' way for the current study is inappropriate and without added value compared to recent work by the co-author team (Hirsch et al., 2017; Hauser et al., 2019) and other researchers (Cook et al., 2011). We therefore downplayed the statements in the manuscript about the potential future impacts of irrigation to highlight these uncertainties:

Our results therefore highlight that irrigation substantially reduced human exposure to warming of hot extremes but question whether this benefit will continue towards the future.

It is unsure whether this evolution will continue in the next decades: besides a possible stagnation (Bonfils and Lobell, 2007; Siebert et al., 2015) or even decrease (Lawrence et al., 2016) in the global area being irrigated, agricultural water use may potentially also become more efficient to meet sustainable development goals related to water resources availability, food security and biodiversity (Jägermeyr et al., 2015, 2017). Even though changes in future irrigation extent and amounts may have important implications in densely-populated irrigation hotspots, current-generation Earth system models generally ignore irrigation in future climate projections (Cook et al., 2011; Hirsch et al., 2017; Hauser et al., 2019). These uncertainties underline the need for including transient irrigation in historical and future climate simulations, and for testing the climate response to various scenarios of future irrigation extent and irrigation efficiency.

Reviewer 3 Comment 5

L29: The words “but only because” seem odd. Why “only”? Is it necessary that the irrigation expanded to say this? It is just because we currently irrigate so much, irrespective of the expansion, no?

Response

We do agree with the reviewer that the wording is a bit odd, so we now simplified the sentence to:

Around one billion people (0.79–1.29) currently benefit from this dampened increase in hot extremes because irrigation massively expanded throughout the 20th century.

Indeed the present-day cooling benefit from irrigation is irrespective of whether irrigation expanded throughout the 20th century or whether it would – hypothetically – have reached its final extent already in, for instance, 1850. However, the point of the current study is that the irrigation expansion *did* parallel the rise in greenhouse gases and that the effect on hot extremes is so strong that it led to a (partial or even complete) offsetting of the signal one would expect to observe or model from climate change alone (Figs. 1-3 in paper). As such, the statement is correct in the sense that it refers to the number of people exposed to ‘dampened increase’ and not to ‘cooling from irrigation’: in case the irrigation expansion would have taken place prior to 1850, present-day cooling would be as strong, but we would also have observed a more pronounced increase in hot extremes during the 20th century.

Reviewer 3 Comment 6

L47: Perhaps a reference to FAO would be more authoritative to state these figures.

Response

We now added a reference to the FAO Aquastat data base. We note however that the Global Map of Irrigated Areas (GMIA) available through FAO is in fact identical to the HID data set we use to compute these numbers.

Reviewer 3 Comment 7

L95-96: This definition is not fully clear here in the text. Why 3-4 days? Why not say “the number of days exceeding the local 99th TX percentile”?

Response

Our goal was to make this variable more comprehensive for the non-expert reader, but we agree that it can be criticised from an expert perspective. We now modified this sentence:

Nearly everywhere across the globe, global warming has increased the likelihood of hot extremes, here defined as the local 99th TX percentile in the early 20th

century (that is, the daytime temperature expected on average once every 100 days) (Fig. 2a).

Reviewer 3 Comment 8

L102: “reverses” may be more appropriate than “offsets” in this case.

Response

Noted and changed.

Reviewer 3 Comment 9

L170: CRU TS is now at version 4.02 while version 3.22 is used here. Could/should the present work be updated, or could this be discarded as irrelevant?

Response

As requested by the reviewer, we updated the observational analysis using the latest version of the CRU data set (v4.02 instead of v3.22). The results are qualitatively consistent, but where the irrigation-induced cooling in v3.22 closely matches the values obtained from CESM (Illustration 8c-d), the results from v4.02 yield a smaller cooling effect (Illustration 7c-d). Overall, this highlights the importance of observational uncertainty, especially if we assume that the main difference between both CRU versions lies in the postprocessing of the raw data (see also Supplementary Information). This strengthens our initial decision to combine both observational data and model simulations. In the revised manuscript, we show the results with v4.02 in the main paper and the results with v3.22 in the SI. We also updated the new Supplementary Fig. 4. See also our response to Reviewer 1 - Comment 3.

Results: While constituting only a small portion of the global land area (0.5%), pixels with an increase in irrigated grid cell fraction above 35% generally experienced a cooling trend. [...] Results of applying the reconstruction method indicate that irrigation expansion has a cooling influence on TXm which increases with greater irrigation extent (Fig. 1c,d), and reaches almost 1 K for the most intensely irrigated regions when using CRU version 3.22 (Supplementary Fig. 2).

Methods: We analyse temperature data from CRU TS v4.02 (hereafter referred to as CRU) and v3.22 (Supplementary Fig. 2), which provides estimates of monthly averaged mean temperature, diurnal temperature range, maximum temperature and minimum temperature at a spatial resolution of $0.5^\circ \times 0.5^\circ$ between 1900 and 2013 (v3.22) or 2017 (v4.02).

Reviewer 3 Comment 10

L172: are all these monthly averages? Or daily?

Response

These are all monthly averages. We rephrased the sentence to make this clear:

We analyse temperature data from CRU TS v4.02 (hereafter referred to as CRU) and v3.22 (Supplementary Fig. 2), which provides monthly averages of mean temperature, diurnal temperature range, daily maximum temperature and daily minimum temperature at a spatial resolution of $0.5^\circ \times 0.5^\circ$ between 1900 and 2013 (v3.22) or 2017 (v4.02).

Reviewer 3 Comment 11

L182: can you provide a reference for this ‘second-order conservative remapping’?

Response

We now added a reference to Jones (1999) in the manuscript.

Reviewer 3 Comment 12

L188-190: How realistic is this of real irrigation practices? Is irrigation in places such as South Asia, particularly in densely irrigated areas such as Punjab, not also limited by restrictions of water use from the regional government?

Response

We investigated the realism of the irrigation parameterisation in our previous study (Thiery et al., 2017) by evaluating simulated irrigation quantities against census data and a suite of observational products. Despite the limitations of the irrigation scheme (see also Reviewer 1 - comment 1), the current CLM4 parameterisation with default settings was found able to realistically represent reported irrigation quantities, notably in South Asia (Illustration 1 and 2; Thiery et al., 2017). Moreover, including irrigation in coupled CESM simulations leads to a small, yet robust increase in the model skill regarding near-surface climate variables (e.g. SEB components, mean and extreme temperatures, precipitation; Thiery et al., 2017). We now included a discussion in the manuscript describing the limitations of irrigation in CLM4 and the skill of the model.

While conducted with a state-of-the-art Earth system model and with great care of achieving realism, our simulations remain characterized by a number of limitations. To start, we assume fixed irrigation extent (of the year 1915 and 2000, respectively) in our otherwise transient simulations as CLM4 cannot handle transient irrigation area. Furthermore, we consider only one crop type (generic C3 crops) and we do not account for variations in irrigation water sources (such as groundwater pumping or rainwater tanks (Demuzere et al., 2014; Leng et al., 2015)), and irrigation techniques (such as sprinkler, drip or flood irrigation (Jägermeyr et al., 2015; Lawston et al., 2015; Leng et al., 2017)). Instead CLM extracts required irrigation amounts from surface runoff and applies it directly to the soil surface. We thus ignore potential local water availability limitations, but note that regional patterns of applied irrigation amounts are realistic when compared to census data (Thiery et al., 2017). Moreover, a set of land-only sensitivity experiments confirms that the default irrigation parameter settings are most suited

for representing irrigation quantities in South Asia (See Supplementary Information).

Reviewer 3 Comment 13

L200: Can you be a bit more specific about how satellite-derived vegetation phenology is imposed in CLM? Is it prescribed as a climatological mean season? But based on what period and what sensor?

Response

Indeed in Satellite Phenology (SP) mode, MODIS-based PFT distributions with matching leaf and stem area index values for are prescribed in the model as a climatological mean season (Lawrence and Chase, 2007). Doing so avoids having to run the model with interactive carbon cycle and associated required millenia-scale spin-up to bring the various carbon pools into equilibrium. As such, running CESM in SP mode greatly reduces computational cost. We now included further explanation on this in the manuscript:

[...] and with satellite-derived vegetation phenology imposed in CLM4.0. The phenology prescription uses leaf area index, stem area index and vegetation height values derived from the Moderate Resolution Imaging Spectroradiometer (MODIS) for the period 2001-2003 and averaged to monthly climatologies at 0.05° spatial resolution (Lawrence and Chase, 2007; Lawrence et al., 2011).

Reviewer 3 Comment 14

L204: Given the large land cover changes that have been occurring during the 20th century, and given that biophysical effects of land cover change such as deforestation have been estimated by co-authors (and other researchers), could this assumption not be alleviated by including these biophysical effects as well?

Response

The decision to keep the land-cover map static in the CESM runs was motivated by the fact that historical land-cover and land management changes (other than irrigation) can have climatic effects opposing or reinforcing those of irrigation, and including them in our simulations would have complicated the retrieval of the temperature effects of irrigation development.

In addition, for the window-searching method, we tested the effect of including a change in tree fraction as a predictor in our results. Since this did not affect our results, we decided to omit this predictor from the final analysis. We thereby note that this result could be anticipated given that deforestation and irrigation typically (though not always) occurred at different locations during the 20th century.

Reviewer 3 Comment 15

L216: Are the spatially moving windows sliding over every pixels are they skipping around to avoid spatial autocorrelation? If the former, what is being done to remove the possible strong spatial autocorrelation?

Response

Our window moves over every pixel, and therefore does not account for auto-correlation. In a recent study using a similar window searching method, Duveiller et al. (2018) use a weighting approach over a larger window to account for auto-correlation. However, given that our study relies on data sets that are an order of magnitude (20x) coarser in terms of spatial resolution, we doubt whether such an approach is appropriate in our case. In addition, we note that the irrigation-induced cooling as obtained from the method demonstrates limited sensitivity the window size when compared to the change in irrigated fraction (i.e. black contours in Illustrations 12 and 8 have a vertical rather than horizontal orientation), suggesting that the influence of auto-correlation on our final results is small. We now added a description of this issue in the method section.

The latest versions (Lejeune et al., 2018; Duveiller et al., 2018) of the method thereby overcome the dependence of an earlier version Kumar et al. (2013) on the binary categorisation of ‘irrigated’ and ‘non-irrigated’ grid cells, by fitting multiple linear regressions between observed temperature changes and changes in irrigation fraction within spatially moving windows.

While a larger search window enables to better capture the temperature contrast between irrigated cells and their surroundings, it also enhances auto-correlation (Duveiller et al., 2018) and spatially smooths local variations of the signal (Supplementary Information). Yet overall the results shown in Fig. 1 demonstrate limited sensitivity to the size of the search window compared to the change in irrigated fraction (Supplementary Figs. 11-12), suggesting that auto-correlation has only limited influence on our results.

Reviewer 3 Comment 16

L216: is the moving window applied to the CRU data at fine spatial resolution or at the regridting resolution of the CESM simulations?

Response

All moving windows analyses were performed at the native CESM resolution ($0.90^\circ \times 1.25^\circ$), i.e. HID and CRU were regridted to the CESM grid using second-order conservative remapping prior to the analysis. We apologise for the confusion in the originally submitted method description, this is now clarified (see our response to Reviewer 3 - Comment 3).

Reviewer 3 Comment 17

L226: it would be welcomed to see this is supplementary material.

Response

This evidence is now provided in the form of two supplementary figures (Illustrations 12 and 13) showing that the sensitivity of the search window size is small compared to the sensitivity to irrigation-induced temperature change in CESM, CRU v3.22 and v4.02. See our response to Reviewer 3 - Comment 3 for a detailed description.

Yet overall the results shown in Fig. 1 demonstrate limited sensitivity to the size of the search window compared to the change in irrigated fraction (Supplementary Figs. 10-11) ...

Reviewer 3 Comment 18

L233: language is a bit too uncertain in: “are likely mostly not accounted for”. How about being more direct: “are not expected to be accounted for by the algorithm” ?

Response

Thank you for this suggestion, which we adopted.

Reviewer 3 Comment 19

L244-249: It would help if you give a clear example of how to interpret the LR metric, e.g.: “a LR of 4 means temperatures above the 99th percentile are 4 times more likely now than before”.

Response

We now included such an example in the method section:

For instance, a LR of 2 and P_{ref} of 0.01 implies that days with a maximum temperature above the 99th percentile are twice as likely in the new period (i.e. a 1-in-50 days event) compared to the reference period (i.e. a 1-in-100 days event).

Reviewer 3 Comment 20

L256-257: phrasing seems to suggest that “irrigated pixels” are not “land pixels”.

Response

We updated this statement:

We compute LR values for land pixels and irrigated land pixels, which are identified as land pixels with an irrigated fraction exceeding 10% of the total grid cell area.

Reviewer 3 Comment 21

L290: typo in author’s initial: A.M.V?

Response

Thank you for noting, indeed this should be A.J.V.

Reviewer 3 Comment 22

change the word ‘cooling’ in the y-axis to ‘change in temperature’ to avoid possible confusion.

Response

This change has been adopted in the revised figure.

Reviewer 3 Comment 23

How do the authors explain the considerable difference between models and observations in the more irrigated pixels?

Response

We believe that uncertainties in both observational data and model simulations are substantial, yet we see no way to determine which uncertainty is largest and therefore responsible for their discrepancy. The observational uncertainty is illustrated when replacing CRU v3.22 by v4.02: while both versions yield qualitatively similar results, magnitudes of irrigation-induced cooling are lower in v4.02 compared to v3.22 (Illustrations 6 and 7). See our response to Reviewer 1 - comment 3 for the changes in the manuscript related to this issue.

References

- Akkermans, T., Thiery, W., and Van Lipzig, N. P. M. (2014). The Regional Climate Impact of a Realistic Future Deforestation Scenario in the Congo Basin. *Journal of Climate*, 27(7):2714–2734.
- Bonfils, C. and Lobell, D. (2007). Empirical evidence for a recent slowdown in irrigation-induced cooling. *Proceedings of the National Academy of Sciences of the United States of America*, 104(34):13582–13587.
- Bright, R. M., Davin, E., O’Halloran, T., Pongratz, J., Zhao, K., and Cescatti, A. (2017). Local temperature response to land cover and management change driven by non-radiative processes. *Nature Climate Change*, 7(4):296–302.
- Cook, B. I., Puma, M. J., and Krakauer, N. Y. (2011). Irrigation induced surface cooling in the context of modern and increased greenhouse gas forcing. *Climate Dynamics*, 37(7-8):1587–1600.
- Cook, B. I., Shukla, S. P., Puma, M. J., and Nazarenko, L. S. (2015). Irrigation as an historical climate forcing. *Climate Dynamics*, 44:1715–1730.
- de Vrese, P., Hagemann, S., and Claussen, M. (2016). Asian irrigation, African rain: Remote impacts of irrigation. *Geophysical Research Letters*, 43(8):3737–3745.

- Demuzere, M., Coutts, A., Göhler, M., Broadbent, A., Wouters, H., van Lipzig, N., and Gebert, L. (2014). The implementation of biofiltration systems, rainwater tanks and urban irrigation in a single-layer urban canopy model. *Urban Climate*, 10:148–170.
- Dirmeyer, P. A. (2011). The terrestrial segment of soil moisture-climate coupling. *Geophysical Research Letters*, 38(16):1–5.
- Dirmeyer, P. A., Wang, Z., Mbuh, M. J., and Norton, H. E. (2014). Intensified land surface control on boundary layer growth in a changing climate. *Geophysical Research Letters*, 41(4):1290–1294.
- Duveiller, G., Hooker, J., and Cescatti, A. (2018). The mark of vegetation change on Earth's surface energy balance. *Nature Communications*, 9(1):679.
- Guimberteau, M., Laval, K., Perrier, A., and Polcher, J. (2012). Global effect of irrigation and its impact on the onset of the Indian summer monsoon. *Climate Dynamics*, 39:1329–1348.
- Guo, Z., Dirmeyer, P. A., Koster, R. D., Bonan, G., Chan, E., Cox, P., Gordon, C. T., Kanae, S., Kowalczyk, E., Lawrence, D., Liu, P., Lu, C.-H., Malyshev, S., Mcavaney, B., Mcgregor, J. L., Mitchell, K., Mocko, D., Oki, T., Oleson, K. W., Pitman, A., Sud, Y. C., Taylor, C. M., Verseghy, D., Vasic, R., Xue, Y., and Yamada, T. (2006). GLACE: The Global Land-Atmosphere Coupling Experiment. Part II: Analysis. *Journal of Hydrometeorology*, pages 611–625.
- Hauser, M., Thiery, W., and Seneviratne, S. I. (2019). Potential of global land water recycling to mitigate local temperature extremes. *Earth System Dynamics*, 10(1):157–169.
- Helkowski, J. (2004). *Global patterns of soil moisture and runoff: An assessment of water availability*. Msc. thesis, University of Wisconsin - Madison, Madison, WI.
- Hirsch, A., Wilhelm, M., Davin, E., Thiery, W., and Seneviratne, S. (2017). Can climate-effective land management reduce regional warming? *Journal of Geophysical Research*, 122(4).
- Hirsch, A. L., Guillod, B. P., Seneviratne, S. I., Beyerle, U., Boysen, L. R., Brovkin, V., Davin, E. L., Doelman, J. C., Kim, H., Mitchell, D. M., Nitta, T., Shiogama, H., Sparrow, S., Stehfest, E., van Vuuren, D. P., and Wilson, S. (2018). Biogeophysical Impacts of Land-Use Change on Climate Extremes in Low-Emission Scenarios: Results From HAPPI-Land. *Earth's Future*, pages 396–409.
- Hooker, J., Duveiller, G., and Cescatti, A. (2018). Data descriptor: A global dataset of air temperature derived from satellite remote sensing and weather stations. *Scientific Data*, 5:1–11.
- Jägermeyr, J., Gerten, D., Heinke, J., Schaphoff, S., Kummu, M., and Lucht, W. (2015). Water savings potentials of irrigation systems: global simulation of processes and linkages. *Hydrology and Earth System Sciences*, 19(7):3073–3091.
- Jägermeyr, J., Pastor, A., Biemans, H., and Gerten, D. (2017). Reconciling irrigated food production with environmental flows for Sustainable Development Goals implementation. *Nature Communications*, 8:15900.

- Jones, P. W. (1999). First- and Second-Order Conservative Remapping Schemes for Grids in Spherical Coordinates. *Monthly Weather Review*, 127(9):2204–2210.
- Keune, J., Sulis, M., Kollet, S., Siebert, S., and Wada, Y. (2018). Human Water Use Impacts on the Strength of the Continental Sink for Atmospheric Water. *Geophysical Research Letters*, 45(9):4068–4076.
- Koster, R. D., Guo, Z., Dirmeyer, P. a., Bonan, G. B., Chan, E., Cox, P. M., Gordon, C. T., Kanae, S., Kowalczyk, E., Lawrence, D. M., Liu, P., Lu, C. H., Malyshev, S., MacAvaney, B., McGregor, J. L., Mitchell, K., Mocko, D., Oki, T., Oleson, K. W., Pitman, A., Sud, Y. C., Taylor, C. M., Verseghy, D., Vasic, R., Xue, Y., and Yamada, T. (2006). GLACE: The Global Land-Atmosphere Coupling Experiment. Part I: Overview. *Journal of Hydrometeorology*, 7:611–625.
- Kueppers, L. M. and Snyder, M. A. (2012). Influence of irrigated agriculture on diurnal surface energy and water fluxes, surface climate, and atmospheric circulation in California. *Climate Dynamics*, 38(5-6):1017–1029.
- Kueppers, L. M., Snyder, M. A., and Sloan, L. C. (2007). Irrigation cooling effect: Regional climate forcing by land-use change. *Geophysical Research Letters*, 34:L03703.
- Kumar, S., Dirmeyer, P. a., Merwade, V., DelSole, T., Adams, J. M., and Niyogi, D. (2013). Land use/cover change impacts in CMIP5 climate simulations: A new methodology and 21st century challenges. *Journal of Geophysical Research: Atmospheres*, 118(12):6337–6353.
- Lawrence, D., Fisher, R., Koven, C., Oleson, K., Swenson, S., Vertenstein, M., Andre, B., Bonan, G., Ghimire, B., van Kampenhout, L., Kennedy, D., Kluzek, E., Knox, R., Lawrence, P., Li, F., Li, H., Lombardozzi, D., and Yaqi, C. X. (2018). CLM5 Documentation. *CLM technical note*.
- Lawrence, D. M., Hurtt, G. C., Arneth, A., Brovkin, V., Calvin, K. V., Jones, A. D., Jones, C. D., Lawrence, P. J., de Noblet-Ducoudré, N., Pongratz, J., Seneviratne, S. I., and Shevliakova, E. (2016). The Land Use Model Intercomparison Project (LUMIP) contribution to CMIP6: Rationale and experimental design. *Geoscientific Model Development*, 9:2973–2998.
- Lawrence, D. M., Oleson, K. W., Flanner, M. G., Thornton, P. E., Swenson, S. C., Lawrence, P. J., Zeng, X., Yang, Z.-L., Levis, S., Sakaguchi, K., Bonan, G. B., and Slater, A. G. (2011). Parameterization improvements and functional and structural advances in Version 4 of the Community Land Model. *Journal of Advances in Modeling Earth Systems*, 3:M03001.
- Lawrence, P. J. and Chase, T. N. (2007). Representing a new MODIS consistent land surface in the Community Land Model (CLM 3.0). *Journal of Geophysical Research*, 112:G01023.
- Lawston, P. M., Santanello, J. A., Zaitchik, B. F., and Rodell, M. (2015). Impact of Irrigation Methods on Land Surface Model Spinup and Initialization of WRF Forecasts. *Journal of Hydrometeorology*, 16:1135–1154.
- Lejeune, Q., Davin, E. L., Gudmundsson, L., Winckler, J., and Seneviratne, S. I. (2018). Historical deforestation locally increased the intensity of hot days in northern mid-latitudes. *Nature Climate Change*, 8(5):386–390.

- Leng, G., Huang, M., Tang, Q., and Lueng, R. (2015). A modeling study of irrigation effects on global surface water and groundwater resources under a changing climate. *Journal of Advances in Modeling Earth Systems*, 7:1285–1304.
- Leng, G., Huang, M., Tang, Q., Sacks, W. J., Lei, H., and Leung, L. R. (2013). Modeling the effects of irrigation on land surface fluxes and states over the conterminous United States: Sensitivity to input data and model parameters. *Journal of Geophysical Research Atmospheres*, 118(17):9789–9803.
- Leng, G., Leung, L. R., and Huang, M. (2017). Significant impacts of irrigation water sources and methods on modeling irrigation effects in the ACME Land Model. *Journal of Advances in Modeling Earth Systems*, 9:1665–1683.
- Lobell, D., Bala, G., Mirin, A., Phillips, T., Maxwell, R., and Rotman, D. (2009). Regional Differences in the Influence of Irrigation on Climate. *Journal of Climate*, 22(8):2248–2255.
- Lobell, D. B., Bonfils, C. J., Kueppers, L. M., and Snyder, M. A. (2008). Irrigation cooling effect on temperature and heat index extremes. *Geophysical Research Letters*, 35(9):1–5.
- Lorenz, R., Pitman, A. J., Hirsch, A. L., and Srbinovsky, J. (2015). Intraseasonal versus Interannual Measures of Land–Atmosphere Coupling Strength in a Global Climate Model: GLACE-1 versus GLACE-CMIP5 Experiments in ACCESS1.3b. *Journal of Hydrometeorology*, 16(5):2276–2295.
- Lu, Y., Jin, J., and Kueppers, L. M. (2015). Crop growth and irrigation interact to influence surface fluxes in a regional climate-cropland model (WRF3.3-CLM4crop). *Climate Dynamics*, 45(11-12):3347–3363.
- Lu, Y. and Kueppers, L. (2015). Increased heat waves with loss of irrigation in the United States. *Environmental Research Letters*, 10(6):064010.
- Luyssaert, S., Jammet, M., Stoy, P. C., Estel, S., Pongratz, J., Ceschia, E., Churkina, G., Don, A., Erb, K., Ferlicoq, M., Gielen, B., Grünwald, T., Houghton, R. A., Klumpp, K., Knohl, A., Kolb, T., Kuemmerle, T., Laurila, T., Lohila, A., Loustau, D., McGrath, M. J., Meyfroidt, P., Moors, E. J., Naudts, K., Novick, K., Otto, J., Pilegaard, K., Pio, C. A., Rambal, S., Reibmann, C., Ryder, J., Suyker, A. E., Varlagin, A., Wattenbach, M., and Dolman, A. J. (2014). Land management and land-cover change have impacts of similar magnitude on surface temperature. *Nature Climate Change*, 4(5):389–393.
- Mueller, B. and Seneviratne, S. I. (2012). Hot days induced by precipitation deficits at the global scale. *Proceedings of the National Academy of Sciences*, 109(31):12398–12403.
- Mueller, N. D., Butler, E. E., Mckinnon, K. A., Rhines, A., Tingley, M., Holbrook, N. M., and Huybers, P. (2016). Cooling of US Midwest summer temperature extremes from cropland intensification. *Nature Climate Change*, 6(March):317–322.
- Oleson, K. W., Lawrence, D. M., Bonan, G. B., Drewniak, B., Huang, M., Charles, D., Levis, S., Li, F., Riley, W. J., Zachary, M., Swenson, S. C., Thornton, P. E., Bozbiyik, A., Fisher, R., Heald, C. L., Kluzek, E., Lamarque, F., Lawrence, P. J., Leung, L. R., Muszala, S., Ricciuto, D. M., and Sacks, W. (2013). Technical description of version 4.5 of the Community Land Model (CLM). *NCAR Technical Note*, 1:420.

- Puma, M. J. and Cook, B. I. (2010). Effects of irrigation on global climate during the 20th century. *Journal of Geophysical Research Atmospheres*, 115(16):1–15.
- Qian, T., Dai, A., Trenberth, K. E., and Oleson, K. W. (2006). Simulation of Global Land Surface Conditions from 1948 to 2004. Part I: Forcing Data and Evaluations. *Journal of Hydrometeorology*, 7(5):953–975.
- Sacks, W. J., Cook, B. I., Buening, N., Levis, S., and Helkowski, J. H. (2009). Effects of global irrigation on the near-surface climate. *Climate Dynamics*, 33:159–175.
- Seneviratne, S. I., Corti, T., Davin, E. L., Hirschi, M., Jaeger, E. B., Lehner, I., Orlowsky, B., and Teuling, A. J. (2010). Investigating soil moisture-climate interactions in a changing climate: A review. *Earth-Science Reviews*, 99(3-4):125–161.
- Seneviratne, S. I., Lüthi, D., Litschi, M., and Schär, C. (2006). Land-atmosphere coupling and climate change in Europe. *Nature*, 443(7108):205–9.
- Seneviratne, S. I., Nicholls, N., Easterling, D., Goodess, C. M., Kanae, S., Kossin, J., Luo, Y., Marengo, J., McInnes, K., Rahimi, M., Reichstein, M., Sorteberg, A., Vera, C., and Zhang, X. (2012). Changes in climate extremes and their impacts on the natural physical environment. In Field, C., Barros, V., Stocker, T., Qin, D., Dokken, D., Ebi, K., Mastrandrea, M., Mach, K., Plattner, G.-K., Allen, S., Tignor, M., and Midgley, P., editors, *Managing the Risks of Extreme Events and Disasters to Advance Climate Change Adaptation*, pages 109–230. Cambridge University Press, Cambridge, UK.
- Seneviratne, S. I., Wilhelm, M., Stanelle, T., Van Den Hurk, B., Hagemann, S., Berg, A., Cheruy, F., Higgins, M. E., Meier, A., Brovkin, V., Claussen, M., Ducharne, A., Dufresne, J. L., Findell, K. L., Ghattas, J., Lawrence, D. M., Malyshev, S., Rummukainen, M., and Smith, B. (2013). Impact of soil moisture-climate feedbacks on CMIP5 projections: First results from the GLACE-CMIP5 experiment. *Geophysical Research Letters*, 40(19):5212–5217.
- Siebert, S., Döll, P., Hoogeveen, J., Faures, J.-M., Frenken, K., and Feick, S. (2005). Development and validation of the global map of irrigation areas. *Hydrology and Earth System Sciences*, 9:535–547.
- Siebert, S., Kummu, M., Porkka, M., Döll, P., Ramankutty, N., and Scanlon, B. R. (2015). A global data set of the extent of irrigated land from 1900 to 2005. *Hydrology and Earth System Sciences*, 19:1521–1545.
- Thiery, W., Davin, E. L., Lawrence, D. M., Hirsch, A. L., Hauser, M., and Seneviratne, S. I. (2017). Present-day irrigation mitigates heat extremes. *Journal of Geophysical Research Atmospheres*, 122:1403–1422.
- Thiery, W., Davin, E. L., Panitz, H.-J., Demuzere, M., Lhermitte, S., and van Lipzig, N. P. M. (2015). The Impact of the African Great Lakes on the Regional Climate. *Journal of Climate*, 28(10):4061–4085.
- Wan, Z. (2014). New refinements and validation of the collection-6 MODIS land-surface temperature/emissivity product. *Remote Sensing of Environment*, 140:36–45.

-
- Winckler, J., Lejeune, Q., Reick, C. H., and Pongratz, J. (2019). Nonlocal Effects Dominate the Global Mean Surface Temperature Response to the Biogeophysical Effects of Deforestation. *Geophysical Research Letters*, 46(2):745–755.

Reviewers' comments:

Reviewer #1 (Remarks to the Author):

The response to the 1st comment of the 1st reviewer is insufficient and carefully avoided to face with the issue how to represent the ponding situation in South Asia that vegetation and water body co-exist at land surface. It is unclear from the Illustration 3 which RMSE change(s) is/are statistically significant and what is the overall improvement by the implementation of IRR scheme.

The authors also avoided to answer directly to the 2nd comment of the 1st reviewer that "Description on how the authors considered the period of irrigation in a year should be included".

Reviewer #2 (Remarks to the Author):

I thank the authors for their elaborate response, which have cleared my concerns. All my comments were addressed, the manuscript was modified correspondingly and additional material and descriptions were added to the Supplement. In particular, I thank the authors for the additional sensitivity analysis of irrigation implementation, which strengthens the results. Congratulations on this interesting work, which unravels an important process affecting temperature extremes. I recommend publication of this paper.

Minor comment: line 217 'in irrigation quantities' instead of 'is irrigation quantities'

Reviewer #3 (Remarks to the Author):

The authors have provided a clear and well-structured response to the initial concerns of the reviewers, and I want to thank them for that. However, some changes are actually causing further concerns at least on one front, which still preclude my recommendation towards publication.

To make bold claims in high impact journals such as Nature Communications based on Earth System Models, it is common to use an ensemble of different models to provide an estimate of the uncertainty related to the models. To strengthen their conclusions, it is better still to combine such study with real observations.

Here the study relies on a single model: CESM. While the authors support that it is adequate to simulate irrigation effects correctly because they have used it effectively in previous studies, it still remains a single model, which limits the possibility to grasp the uncertainty around its results. Some of the possible shortcomings were already pointed out by reviewer 1. I fully understand that running various ESMs here is well beyond the scope of the work, that it will probably be done in future work by the authors. This is probably the first motivation for the authors to have used the CRU dataset as a source of additional observational evidence, which is clearly welcome. However, the newly presented results using the updated version of CRU (as requested in the first round of reviewing) shows quite surprising results. If I understand correctly from Figure 1, Figure S4 and Figure S11 & S12, the CRU v4.02 shows a much weaker signal than CRU v3.22, which itself is already considerably weaker than what CESM simulates. This is especially true in South Asia, which is a focal point in the main message of the paper. While the direction of the signal is consistent (e.g. Figure 1 c and d), the magnitude is very different, which casts some doubts on whether the conclusions based on the model only (e.g. Figure 2 and 3) are not also magnified due to shortcomings of the model. How to solve that?

First, as the CRU v4.02 has super-seeded previous versions, I believe there should be no reference to the deprecated version at all in the manuscript nor in the supplementary material. I understand there might be a temptation to keep results with v3.22 since they show closer agreement with the CESM results (especially in the spatial patterns in Figure S4 for instance), but this will be perceived

negatively by readers. It is my understanding that newer versions of CRU should be better and more realistic as incremental improvements are incorporated. Keeping the single latest version seems the best option to avoid any confusion from the reader.

With the old CRU out and the wider disagreement between these 'observation' results and the CESM simulations, it becomes even more important (in my opinion) to bring something more to ensure the robustness of the main tool (CESM) and to support the results if the authors insist in publishing in a high impact journal such as Nature Communication. As I mentioned in my first review, I myself have doubts on whether the gridding CRU dataset would be the right 'observational' data to use (Reviewer 3 Comment 2 in the response to reviewers), and I suggested to consider satellite observations. The authors judiciously countered this argument by noting that their main objective is to see how the effect of irrigation evolved in time rather than exploring the present-day spatio-temporal patterns, thereby ruling out remote sensing. The authors also countered my other suggestion of using the sub-grid CESM results instead of 'unmixing' gridded simulations (Reviewer 3 Comment 3) by noting that air temperature, the variable of interest, is only available at grid scale. Actually, I think a combination of these two propositions may provide the needed support to consolidate trust in the CESM simulations.

My suggestion would be to compare differences between irrigated and non-irrigated sub-grid estimates of CESM surface temperature (which do exist according to the authors) with those obtained from remote sensing from instruments like MODIS over nearby irrigated and non-irrigated pixel during the 'present' era (e.g. 2000-2010 to match the time horizon of your current CESM simulations, but perhaps the actual years do not matter that much). Perhaps this analysis could even be limited to South Asia if necessary, as it is in the centre of the message of the manuscript. First, this analysis should bring insight on the skill of the model to represent the present-day spatio-temporal patterns of irrigated vs non-irrigated maps at sub-grid scale, thereby allowing to have an idea of the possible biases it may have on the magnitude of the irrigation effect on LST. Of course, this will be done on LST and not air temperature, and there may still be some uncertainty. But if CESM catches the LST right, it should be able to internally translate it to air temperature reasonably well. Second, this analysis will also provide insight on the robustness of the unmixing window technique for this specific context of irrigated crops (which is arguably different from the one of Lejeune et al. 2017 in forests with larger pixels). Overall this analysis, which could be left mostly in supplementary material if so desired, would ensure the due credibility on the skill of CESM to perform the task and thus reinforce the message of the paper. Technical point: when you do this analysis, do filter to select only 'clear-sky' days in the model output to make it comparable to the satellite measurements (which are only done when there are no clouds in the way).

Further, more punctual, remarks:

Line 76: In my opinion, the reference to CRU v3.22 and to any results derived from it should be removed from the main text, as this dataset is superseded by the latter one. Even in supplementary material, it is questionable to keep it there. A reader wonders why that particular version should be there.

Line 103: actually, why is the Sahara blue? Why would there be any irrigation effect if no irrigation can be reasonably attributed there?.

Line 184: double 'that'

Lines 227-238: this paragraph should also include the limitation of using a prescribed seasonal LAI in the model, resulting in a mismatch with reality. This is important as LAI is a key variable describing the interface between plants and atmosphere. In case of drought, for instance, while irrigated crops may not be affected much (as one would assume irrigation is sufficient to ensure proper 'normal' growth of LAI), the non-irrigated crops around that serve as comparison would be wrongly modelled as having a high LAI.

Figure 2 caption should somehow specify that the data comes from model simulations (even if obvious for the authors and insiders, it may not be so for more casual readers).

Figure 3: just to check, the bars don't add up as one could expect intuitively. I suppose this is because, as you mention in lines 300-301, they are based on different ensembles. I suggest you introduce that in the caption to avoid confusion

Figure S6: what is the source of this data? Are these model simulated results? Or actual/real estimates?

SI Line 37: "Interestingly, (i) CESM has less sensitivity compared to CRU"... I do not understand. I see the opposite in Figure S11. Values for CESM are mostly higher and more variable. Furthermore, could you please elaborate on why you think the sensitivity reported in Figure S11 is so much higher and variable in CESM than in CRU v4.02.

Warming of hot extremes alleviated by expanding irrigation

Submitted to Nature Communications

August 7, 2019

Wim Thiery^{1, 2}, Auke J. Visser³, Erich M. Fischer¹, Mathias Hauser¹, Annette L. Hirsch⁴,
David M. Lawrence⁵, Quentin Lejeune⁶, Edouard L. Davin¹, Sonia I. Seneviratne¹
wim.thiery@env.ethz.ch

¹ETH Zurich, Institute for Atmospheric and Climate Science, Universitaetsstrasse 16, 8092 Zurich, Switzerland.

²Vrije Universiteit Brussel, Department of Hydrology and Hydraulic Engineering, Pleinlaan 2, 1050 Brussels, Belgium.

³Wageningen University, Meteorology and Air Quality group, Droevendaalsesteeg 3a, 6708PB, Wageningen, the Netherlands.

⁴University of New South Wales, ARC Centre of Excellence for Climate Extremes, 2052 Sydney, Australia.

⁵National Center for Atmospheric Research, Boulder, Colorado, USA.

⁶Climate Analytics, Ritterstrasse 3, 10969 Berlin, Germany.

Contents

1	Reviewer 1	3
	Reviewer 1 Comment 1	3
	Reviewer 1 Comment 2	4
	Reviewer 1 Comment 3	4
2	Reviewer 2	7
	Reviewer 2 Comment 1	7
	Reviewer 2 Comment 2	7
3	Reviewer 3	8
	Reviewer 3 Comment 1	8
	Reviewer 3 Comment 2	8
	Reviewer 3 Comment 3	8
	Reviewer 3 Comment 4	9
	Reviewer 3 Comment 5	16
	Reviewer 3 Comment 6	17
	Reviewer 3 Comment 7	17
	Reviewer 3 Comment 8	17
	Reviewer 3 Comment 9	18
	Reviewer 3 Comment 10	18

Reviewer 3 Comment 11	18
Reviewer 3 Comment 12	19

Abstract

We would like to once again thank all reviewers for their dedicated time reviewing the response letter and revised manuscript, and for their useful and constructive suggestions in this new round. We carefully addressed all comments by the reviewers. Next to the modifications detailed below, we note that we replaced all mentions of 'likelihood ratio' (LR) to 'probability ratio' (PR) to be consistent with Fischer and Knutti (2015).

1 Reviewer 1

Reviewer 1 Comment 1

The response to the 1st comment of the 1st reviewer is insufficient and carefully avoided to face with the issue how to represent the ponding situation in South Asia that vegetation and water body co-exist at land surface.

Response

We agree with Reviewer 1 that ponding is a widely used irrigation technique in South Asia and that its current representation in CESM as 'normal' irrigation represents a shortcoming of the study. More generally, the same critique applies to other techniques and regions, for instance, in some places very efficient drip irrigation techniques are used which are likewise not represented in CESM. Yet so far there is - to our knowledge - not a single global Earth system model that represents multiple irrigation techniques and ponding irrigation in particular. In fact, all CMIP5 models completely ignored irrigation, and many CMIP6 models still do or have an irrigation parameterisation that is at best comparable to the one in CESM. Including a parameterisation of ponding irrigation in a climate model requires extensive theoretical work and programming, searching for and integrating of input data sets (e.g. global maps of irrigation types), parameter sensitivity testing and model evaluation. This fall a student will start his PhD in the group of the first author to work exactly on implementing various irrigation techniques in CLM during the next four years. Moreover, paddy rice has been implemented very recently in a regional version of CLM4 (Devanand et al., 2019) based on earlier developments in SWAT and MATSIRO (Masutomi et al., 2016; Xie and Cui, 2011). Therefore we trust that this functionality will become available in CESM in the near future, and hope that the reviewer understands that it is beyond the scope of the current study to implement ponding irrigation in CESM.

In response to this feedback, we have now moved the limitations paragraph from the methods to the main text, reworked it and extended it with a literature review of ponding irrigation effects on climate:

While conducted with a state-of-the-art Earth system model and with great care of achieving realism, our simulations remain characterized by a number of limitations. To start, we assume fixed irrigation extent (of the year 1915 and 2000, respectively) in our otherwise transient simulations as the current model version cannot handle transient irrigation area. Furthermore, we consider only one crop type (generic C3 crops) and we do not account for variations in irrigation water sources (such as groundwater pumping or rainwater tanks (Demuzere et al., 2014; Leng et al., 2015)), and irrigation techniques (such as sprinkler, drip, flood or ponding irrigation (Jägermeyr et al., 2015; Lawston et al., 2015; Leng et al., 2017; Devanand et al., 2019; Masutomi et al., 2016; Xie and Cui, 2011)). Instead CESM extracts required irrigation amounts from surface runoff and applies it directly to the soil surface, where it can either infiltrate, evaporate or run off. We thus ignore potential local water availability limitations, but note that regional patterns of applied irrigation amounts are realistic when compared to census data (Thiery et al., 2017). This is confirmed by a set of land-only sensitivity ex-

periments showing that the default irrigation parameter settings are most suited for representing irrigation quantities in South Asia (See Supplementary Information). However, even though our simulated irrigation quantities closely match observed values over South Asia (Thiery et al., 2017), water ponding is currently disabled in CESM, which likely leads to an underestimation of irrigation impacts on temperature in South, East and Southeast Asia where paddy fields are widespread (Devanand et al., 2019). Comparison of simulated and satellite based land surface temperatures indeed confirms that CESM appears to underestimate irrigation effects on present-day land surface temperatures across South Asia (see Supplementary Information). Implementing paddy irrigation in CESM would thus be beneficial, especially since observational studies (Liu et al., 2019b,a; Chen and Jeong, 2018) demonstrate that paddy field expansion in China may locally lead to land surface temperature reductions on the order of 1-2 K on average during the growing season.

Reviewer 1 Comment 2

It is unclear from the Illustration 3 which RMSE change(s) is/are statistically significant and what is the overall improvement by the implementation of IRR scheme.

Response

We have now updated illustration 3 of the previous version to indicate which temporal RMSE changes are statistically significant using the paired, two-sided Wilcoxon signed rank test (illustration 1; significant changes at $p < 0.05$ are marked by a grey dot). Note that the colors denote the change in spatiotemporal RMSE and thus compare two numbers. As it makes no sense to compute a test statistic on two values, the statistical significance was instead computed for the temporal RMSE (i.e. the spatially-explicit signal of annual RMSE computed per pixel based on multi-year monthly means).

In general, spatiotemporal RMSE values decrease for 77 out of a total of 100 considered samples (77%), with an average RMSE decrease of 3%. Over irrigated land, an improvement is found for all considered observational products except GLEAM, whereas over global land the model skill improves for all variables except TNn (Figure 1). Thus, while the enhanced skill is small in absolute terms, it is consistent across a wide range of climatic variables and relevant spatial domains. The statistical test applied to the temporal RMSE values confirms that differences are statistically significant in most cases. We therefore conclude that irrigation has a small yet overall beneficial effect on the representation of present-day near-surface climate (Thiery et al., 2017).

Reviewer 1 Comment 3

The authors also avoided to answer directly to the 2nd comment of the 1st reviewer that "Description on how the authors considered the period of irrigation in a year should be included".

Response

Irrigation can only happen during the C3 crop growing season, i.e. only when $LAI_{C3} > 0$,

[redacted]

Illustration 2: Seasonal cycle of irrigation quantities as simulated by CESM for the IRR ensemble.

whereby LAI values are prescribed from satellite observations. Yet within this period we do not prescribe when the model actually performs irrigation. Instead, CLM itself computes per day if (and how much) irrigation needs to happen based on plant water stress. This results in crops requiring irrigation during dry periods and not requiring irrigation during wet periods of the growing season.

In terms of irrigation seasonality, this results in most regions receiving (in the model) maximum irrigation amounts during boreal summer months June to August (MED, WAS, WNA, CNA, EAS), whereas the southern Asian Monsoon regions receive most irrigation during the months January to March (SEA) and March to May (SAS; Illustration 2).

We updated the manuscript to include a discussion of how irrigation is triggered in the model and what the resulting irrigation seasonality is:

Whenever soil moisture is limiting photosynthesis during the growing season, irrigation is activated, and the deficit between the actual and a target soil moisture content applied to the ground surface in a way that mimicks extraction from nearby rivers. Although confined to the crop growing season, timing and quantities of irrigation are not prescribed, but internally computed by the irrigation paramaterisation in CLM (Lawrence et al., 2011; Thiery et al., 2017). In terms of irrigation seasonality, this results in most irrigation hotspots receiving maximum irrigation amounts during boreal summer months, whereas South and Southeast Asia regions receive most irrigation during boreal spring.

2 Reviewer 2

Reviewer 2 Comment 1

I thank the authors for their elaborate response, which have cleared my concerns. All my comments were addressed, the manuscript was modified correspondingly and additional material and descriptions were added to the Supplement. In particular, I thank the authors for the additional sensitivity analysis of irrigation implementation, which strengthens the results. Congratulations on this interesting work, which unravels an important process affecting temperature extremes. I recommend publication of this paper.

Response

We thank Reviewer 2 for their time revising the manuscript and response letter, and for the recommendation to publish this study.

Reviewer 2 Comment 2

Minor comment: line 217 'in irrigation quantities' instead of 'is irrigation quantities'.

Response

This typo has been corrected in the revised manuscript.

3 Reviewer 3

Reviewer 3 Comment 1

The authors have provided a clear and well-structured response to the initial concerns of the reviewers, and I want to thank them for that. However, some changes are actually causing further concerns at least on one front, which still preclude my recommendation towards publication.

Response

We thank the Reviewer for the appreciation of the response letter and revisions we implemented in the original manuscript. Below, we address the issues raised in this new review round.

Reviewer 3 Comment 2

To make bold claims in high impact journals such as Nature Communications based on Earth System Models, it is common to use an ensemble of different models to provide an estimate of the uncertainty related to the models. To strengthen their conclusions, it is better still to combine such study with real observations.

Here the study relies on a single model: CESM. While the authors support that it is adequate to simulate irrigation effects correctly because they have used it effectively in previous studies, it still remains a single model, which limits the possibility to grasp the uncertainty around its results. Some of the possible shortcomings were already pointed out by reviewer 1. I fully understand that running various ESMs here is well beyond the scope of the work, that it will probably be done in future work by the authors. This is probably the first motivation for the authors to have used the CRU dataset as a source of additional observational evidence, which is clearly welcome.

Response

We thank the reviewer for their understanding regarding climate modellers being tied to a single ESM for conducting global-scale experiments. Indeed, including observational analyses to supplement the model experiments was motivated by our attempt to mitigate this limitation.

Reviewer 3 Comment 3

However, the newly presented results using the updated version of CRU (as requested in the first round of reviewing) shows quite surprising results. If I understand correctly from Figure 1, Figure S4 and Figure S11 and S12, the CRU v4.02 shows a much weaker signal than CRU v3.22, which itself is already considerably weaker than what CESM simulates. This is especially true in South Asia, which is a focal point in the main message of the paper. While the direction of the signal is consistent (e.g. Figure 1 c and d), the magnitude is very different, which casts some doubts on whether the conclusions based on the model only (e.g. Figure 2 and 3) are not also magnified due to shortcomings of the model. How to solve that? First, as the CRU v4.02 has superseded previous versions, I believe there should be no reference to the deprecated version at all in the manuscript nor in the supplementary material. I understand there might be a temptation to keep results with v3.22 since they show closer agreement with the CESM results (especially in the spatial patterns in Figure S4 for instance), but this will be perceived negatively by readers. It is my understanding that newer versions of CRU should be better and more realistic as incremental improvements are incorporated. Keeping the single latest version seems the best option to avoid any confusion from the reader.

Response

The reviewer is correct in observing that the newer CRU version (v4.02) shows a weaker signal compared to the version used in the original submission (v3.22). For us, the large difference between both versions of the same observational data set is a clear sign that the observational data set is characterised by uncertainty, just like the model simulations are. This is confirmed by the fact that the main difference between both CRU versions is a new interpolation algorithm, not an increase in the number of stations included in the data product. Moreover, the new analysis on CESM and MODIS land surface temperatures (see below) suggests that CESM underestimates rather than overestimates the MODIS-based cooling estimate, confirming that both models and observational products need to be treated with caution here. That said, we followed the reviewer's recommendation and removed all figures of and references to the old CRU version in the revised manuscript and SI.

Reviewer 3 Comment 4

With the old CRU out and the wider disagreement between these ‘observation’ results and the CESM simulations, it becomes even more important (in my opinion) to bring something more to ensure the robustness of the main tool (CESM) and to support the results if the authors insist in publishing in a high impact journal such as Nature Communication. As I mentioned in my first review, I myself have doubts on whether the gridding CRU dataset would be the right ‘observational’ data to use (Reviewer 3 Comment 2 in the response to reviewers), and I suggested to consider satellite observations. The authors judiciously countered this argument by noting that their main objective is to see how the effect of irrigation evolved in time rather than exploring the present-day spatio-temporal patterns, thereby ruling out remote sensing. The authors also countered my other suggestion of using the sub-grid CESM results instead of ‘unmixing’ gridded simulations (Reviewer 3 Comment 3) by noting that air temperature, the variable of interest, is only available at grid scale. Actually, I think a combination of these two propositions may provide the needed support to consolidate trust in the CESM simulations. My suggestion would be to compare differences between irrigated and non-irrigated sub-grid estimates of CESM surface temperature (which do exist according to the authors) with those obtained from remote sensing from instruments like MODIS over nearby irrigated and non-irrigated pixel during the ‘present’ era (e.g. 2000-2010 to match the time horizon of your current CESM simulations, but perhaps the actual years do not matter that much). Perhaps this analysis could even be limited to South Asia if necessary, as it is in the centre of the message of the manuscript. First, this analysis should bring insight on the skill of the model to represent the present-day spatio-temporal patterns of irrigated vs non-irrigated maps at sub-grid scale, thereby allowing to have an idea of the possible biases it may have on the magnitude of the irrigation effect on LST. Of course, this will be done on LST and not air temperature, and there may still be some uncertainty. But if CESM catches the LST right, it should be able to internally translate it to air temperature reasonably well. Second, this analysis will also provide insight on the robustness of the unmixing window technique for this specific context of irrigated crops (which is arguably different from the one of Lejeune et al. 2017 in forests with larger pixels). Overall this analysis, which could be left mostly in supplementary material if so desired, would ensure the due credibility on the skill of CESM to perform the task and thus reinforce the message of the paper. Technical point: when you do this analysis, do filter to select only ‘clear-sky’ days in the model output to make it comparable to the satellite measurements (which are only done when there are no clouds in the way)

Response

As requested by the reviewer, we have now included an analysis of irrigation effects in remotely sensed temperatures and subgrid-scale model output to test the quality of CESM and the robustness of the the unmixing algorithm.

Introduction. To test the quality of CESM and the robustness of the window searching algorithm, we applied the algorithm to land surface temperatures (LST) from MODIS and the CESM IRR ensemble. We note that there are some fundamental differences between the

CESM comparison to CRU (hereafter referred to as the CRU analysis) and the new analysis on CESM and MODIS (hereafter referred to as the MODIS analysis):

1. When analysing MODIS, we consider only the spatial signal, not the spatiotemporal signal, that is, we look at the difference in present-day absolute LST between irrigated and surrounding non-irrigated regions, rather than at how the local temporal change signal differs between irrigated and surrounding non-irrigated regions (as in the CRU analysis).
2. The MODIS analysis considers LST instead of near-surface air temperatures (T).
3. We perform the MODIS analysis both on the grid-scale and subgrid scale model output, whereas the CRU analysis is performed at the grid-scale only (this is justified by the resolution of CRU being much coarser compared to MODIS). See below for a detailed description of the calculations on the subgrid-scale output.
4. The MODIS grid-scale analysis and CRU analysis are performed at the CESM resolution, while the MODIS subgrid-scale analysis is performed at the spatial resolution of the Historical Irrigation Database (HID).
5. While daily maximum 2m air temperatures can be analysed in CESM, for LST we can only analyse 24h averages from the model due to raw output data limitations.

Methods. The average skin temperatures are calculated for MODIS (based on monthly data from MODIS/Aqua MYD11C3-v006 available at 0.05° spatial resolution (Wan, 2008)) and CESM for 2003-2010, i.e. the years in the MODIS data record with a full annual cycle that overlap with the CESM simulations. In the following, we separate the MODIS analysis into a grid-scale and a subgrid-scale analysis.

Grid-scale analysis: The grid-scale analyses have been performed using a modified version of the window searching algorithm (see Methods section in the main text). We perform the regression on the present-day total LST signal for the months March-May (MAM), the dry pre-monsoon season in South Asia. Since this analysis concerns spatial irrigation impacts, we use the present-day irrigated fraction derived from HID as a regressor (see Methods Equation 1). The model results are calculated from LST output, which was computed from the raw LW_{out} output using the Stefan-Boltzmann law. We use MODIS/Aqua observations from the 01:30 h LT and 13:30 h LT overpass (hereafter referred to as nighttime and daytime overpass, respectively) for the calculation of the observed irrigation impacts on LST. These data were regridded to the CESM-grid using second-order conservative remapping (Jones, 1999) before application in the search window algorithm.

Subgrid-scale analysis: the high resolution of the MODIS observations enables a quantification of the irrigation-induced LST change at a finer resolution than the model output. The sub-grid effect is calculated by taking the difference in skin temperature between pairs of irrigated and non-irrigated crop tiles, to calculate the 'potential irrigation-induced cooling' between a fully irrigated pixel and a non-irrigated pixel. This is then multiplied by the irrigated fraction to obtain an estimate of the actual irrigation-induced cooling rate:

$$\Delta LST_{irr} = f_{irr} \times (LST_{crop,irrigated} - LST_{crop,rainfed})$$

The sub-grid irrigation effect in MODIS is calculated by application of the search window algorithm to MODIS LST observations (mean of daytime and nighttime overpass when compared to CESM, daytime overpass only when analysed separately), regridded to a 5 arc min resolution of the HID grid. The algorithm configuration is equivalent to that of the grid-scale analysis, but a search window size of 13×13 cells is used to obtain a total search window size of $1.08^\circ \times 1.08^\circ$, so that a search window closely resembles the size of a CESM pixel ($1.25^\circ \times 0.9375^\circ$). The subgrid-scale calculations are performed on annual average PFT-level CESM output and average daytime and nighttime MODIS overpasses over the period 2003–2010.

Finally, we note that three caveats imply that there will be an inherent difference between the MODIS- and CESM-based results:

1. Both LST and cloud cover are stored only at monthly time step for our CESM runs. Hence we cannot exclude cloudy days from the CESM output. As such, the MODIS data contains only clear-sky values whereas the CESM output contains all-sky values.
2. We assume that emissivity equals 1 when converting the raw CESM LW_{out} values to LST using the Stefan-Boltzmann law. In reality, emissivity may differ from 1. Moreover, the resulting LST values are sensitive to the choice of the emissivity value.
3. For MODIS, we take the monthly mean of the average of the daytime and nighttime overpasses, which correspond more or less to the daily maximum and daily minimum, respectively, whereas for CESM we can only work with the monthly mean of the 24h average as the raw LW_{out} data is only stored at this temporal resolution.

This discrepancy between MODIS and CESM is at least partly because the simulations (and their output) were originally not designed towards this comparison (see e.g. the suggestion by the reviewer to apply cloud masking). Hence we include the full results in the SI of the paper and summarise them in the main text, as they further show that CESM is a good tool to study climate impacts of irrigation.

Results. The absolute LST values as simulated by CESM correspond reasonably well to MODIS values (Fig. 3a). A cold bias during the winter months is consistent with two inherent differences between CESM and MODIS listed above: no cloud masking and an emissivity of 1 both tend towards a cold bias in CESM. Given the current model set-up it is however difficult to determine which of these effects is the dominant factor in explaining biases in CESM.

Application of the spatial window-searching method to CESM and MODIS LST (grid-scale) indicates that CESM underestimates the irrigation-induced cooling relative to MODIS (Illustration 3b). We note that this is opposite to the response obtained from applying the spatio-temporal window-searching method to CESM CRU T (grid-scale), where CESM overestimates the irrigation signal compared to CRU (Figure 1). Overall this suggests the presence of substantial uncertainties in both model and observational products, but also highlights differences in the response of LST versus T to irrigation. The latter is consistent with recent results for another land surface forcing (Winckler et al., 2019) showing that both in models and observations, near-surface air temperatures respond very differently to deforestation compared to land surface temperatures. Despite the uncertainty in the magnitude of the cooling, the signal of the irrigation-induced effect on LST is remarkably consistent between

Illustration 3: Observed and simulated land surface temperatures affected by irrigation. Seasonal cycle in observed and simulated absolute land surface temperatures (LST) across South Asia (a), irrigation-induced change in present-day absolute LST as a function of present-day irrigated fraction (b), and scatter plot of present-day simulated versus observed irrigation-induced change in LST (c). The window searching algorithm was applied during the months March-May (MAM; grey band in panel a). Blue lines and boxplots represent results from the MODIS observational data, orange lines and boxplots show results from the CESM global climate simulations under present-day climate conditions (grid-scale analysis). Boxplots indicate the spatial distribution (center line: median; box limits: upper and lower quartiles; whiskers and outliers: not shown).

CESM and MODIS in terms of sign (Illustration 3c): only 18 out of 326 considered pixels show an opposite sign in their response to irrigation (5.52%). Illustration 3c furthermore shows that both in CESM and MODIS the strongest cooling occurs for pixels with the highest present-day irrigated fraction. From these results we conclude that CESM may be used a tool to investigate irrigation effects on surface climate, but that absolute values need to be interpreted with caution.

The irrigation-induced cooling during daytime as retrieved from MODIS for South Asia has a marked seasonality, with the strongest cooling during the months March-May (MAM) and hardly any effect during June-August (JJA; Illustration 4b). This is consistent with Illustration 5 adapted from Thiery et al. (2017) showing the strongest cooling effect for LST in CESM during MAM and a rapid removal of the signal around the onset of the Indian Monsoon. This reinforces our earlier conclusion that CESM may be used in this context.

Finally, comparison of the MODIS window-searching method to the CESM subgrid-scale effect shows a relatively consistent median response in model and observations over South Asia but a larger spatial variability in MODIS (Illustration 6). Probably this difference in variability is owing to the large difference in spatial resolution between CESM and MODIS.

Discussion. The above analysis highlights the challenges of comparing models and observation characterised by uncertainties, but at the same time demonstrates that CESM may be used as a tool for robust irrigation impact assessments. Furthermore, we believe that it is difficult to compare irrigation-induced changes in T and LST. This is consistent with recent findings (Winckler et al., 2019) that LST and T may respond differently to deforestation in Earth system models. Consequently, deriving irrigation-induced changes in T based on LST is not straightforward, and we have therefore not attempted such a conversion here.

Illustration 4: Seasonal signature of irrigation influence on observed land surface temperatures. Irrigation-induced change in present-day absolute land surface temperature (LST) as a function of present-day irrigated fraction for December-January (DJF) (a), March-May (MAM) (b), June-August (JJA) (c), and September-November (SON) (d). Blue bars represent results for irrigated land, whereas orange bars show results for South Asia. Irrigated land is defined here as all pixels with >10% irrigated crop fraction, and corresponds to ~5% of all land area. South Asia is defined as Pakistan, India, Nepal and Bangladesh, and represents ~3% of all land. Boxplots indicate the spatial distribution (center line: median; box limits: upper and lower quartiles; whiskers and outliers: not shown).

[redacted]

Illustration 6: **Comparison of irrigation effects on land surface temperatures as derived from subgrid-scale CESM output and MODIS data.** Irrigation-induced change in land surface temperature (LST) across South Asia as derived from applying the window-searching method to MODIS (blue) and from scaling the subgrid-scale LST difference between irrigated and rainfed crop tiles by the present-day irrigated fraction (orange, see Supplementary Information text). Results are representative of annual conditions during 2003-2010. Boxplots indicate the spatial distribution (center line: median; box limits: upper and lower quartiles; whiskers and outliers: not shown).

Future research may however further investigate the potential of remote sensing to derive changes in near-surface air temperature from LST data (Hooker et al., 2018), and explore the potential of using subgrid-scale output from multiple Earth system models to derive robust biogeophysical responses to land cover changes.

We have now added a new section in the revised SI describing this new analysis and included three new figures (Illustrations 3, 4, and 6). We have also updated the main text to include a summary of the analysis:

Although the irrigation-induced cooling in monthly temperature extremes is larger in CESM than in the observation-based estimate, the two lines of evidence yield consistent results (Fig. 1, Supplementary Fig. 4) and are corroborated by the comparison of simulated and satellite-based land surface temperatures showing consistent and substantial irrigation-induced cooling in intensely irrigated regions (Supplementary Information). These analyses reaffirm earlier conclusions (Thiery et al., 2017) that the model can be used to study the effects of irrigation on climate extremes, while the exact magnitude of the effect needs to be interpreted with caution.

However, even though our simulated irrigation quantities closely match observed values over South Asia (Thiery et al., 2017), water ponding is currently disabled in CESM, which likely leads to an underestimation of irrigation impacts on temperature in South, East and Southeast Asia where paddy fields are widespread (Devanand et al., 2019). Comparison of simulated and satellite based land surface temperatures indeed confirms that CESM appears to underestimate irrigation effects on present-day land surface temperatures across South Asia (see Supplementary Information). Implementing paddy irrigation in CESM would thus be beneficial, especially since observational studies (Liu et al., 2019b,a; Chen and Jeong, 2018) demonstrate that paddy field expansion in China may locally lead to land surface temperature reductions on the order of 1-2 K on average during the growing season.

Note that the long time span of the analysis period impedes the use of satellite-based surface or air temperature data sets (Wan, 2008; Hooker et al., 2018), which have recently been shown to be powerful tools for assessing the local biophysical effects of land cover changes (Bright et al., 2017; Duveiller et al., 2018). We therefore use satellite data only for assessing present-day effects of irrigation (see Supplementary Information).

Reviewer 3 Comment 5

Line 76: In my opinion, the reference to CRU v3.22 and to any results derived from it should be removed from the main text, as this dataset is superseded by the latter one. Even in supplementary material, it is questionable to keep it there. A reader wonders why that particular version should be there.

Response

See our response to Reviewer 3 - Comment 3: We have removed all references to CRU

v3.22.

Reviewer 3 Comment 6

Line 103: actually, why is the Sahara blue? Why would there be any irrigation effect if no irrigation can be reasonably attributed there?.

Response

The reduced likelihood of the 99th percentile TX due to irrigation is consistent with the cooling of annual maximum temperature (TXx) reported by Thiery et al. (2017) for CESM. Interestingly, the model simultaneously suggests a warming of the annual mean (T) and annual minimum (TNn) temperatures. Analysis of the surface energy balance components suggests that over the Sahara (and according to CESM), a slight cooling occurs during JJA due to increased latent heat flux combined with reduced incoming longwave radiation, although the magnitude of both effects is small (on the order of 0.1°C). However, other model studies show limited effects over the Sahara (e.g. Cook et al., 2015; de Vrese et al., 2016). A review of the literature on climatic impacts of irrigation overall shows that many studies report remote effects, but that these remote effects often seem to be model dependent (e.g. strong precipitation changes in the Amazon in CESM (Thiery et al., 2017) versus strong changes in East Africa in MPI-ESM, (de Vrese et al., 2016)). For that reason, we think that remote effects of irrigation need to be treated with great care and preferably investigated in a multi-model setting (exactly what we plan to do within a new research project starting September 2019).

Reviewer 3 Comment 7

Line 184: double 'that'.

Response

This typo has been corrected in the revised manuscript.

Reviewer 3 Comment 8

Lines 227-238: this paragraph should also include the limitation of using a prescribed seasonal LAI in the model, resulting in a mismatch with reality. This is important as LAI is a key variable describing the interface between plants and atmosphere. In case of drought, for instance, while irrigated crops may not be affected much (as one would assume irrigation is sufficient to ensure proper 'normal' growth of LAI), the non-irrigated crops around that serve as comparison would be wrongly modelled as having a high LAI.

Response

Good point. It is one of the reasons why the irrigation module has been incorporated in the crop model in subsequent releases of CLM, which implies that irrigation can only be run together with an interactive carbon cycle. While this substantially enhances both spin-up requirements and associated computational costs, it increases the realism of crop response to irrigation. We will include this interaction in future studies, but at the time the current

simulations were set up it was unfortunately not a feasible option.

We adopted the paragraph accordingly and moved it from the methods to the main text (see our response to Reviewer 1 - Comment 1):

Furthermore, we consider only one crop type (generic C3 crops) with prescribed leaf area index (that is, crop phenology does not interact with irrigation; see Methods) ...

Reviewer 3 Comment 9

Figure 2 caption should somehow specify that the data comes from model simulations (even if obvious for the authors and insiders, it may not be so for more casual readers).

Response

Good point, we updated the figure caption:

Ensemble-mean likelihood of exceeding 99th percentile of daytime temperature TX as simulated by CESM,...

Reviewer 3 Comment 10

Figure 3: just to check, the bars don't add up as one could expect intuitively. I suppose this is because, as you mention in lines 300-301, they are based on different ensembles. I suggest you introduce that in the caption to avoid confusion.

Response

Indeed the bars don't add up because the reference for the dark blue bars (IRR) is different from the reference for the light blue and red bars (IRR_20C). We updated the caption of figure 3 to clarify this point:

Note that the bars are non-additive because of differences in the reference ensemble (see Methods).

Reviewer 3 Comment 11

Figure S6: what is the source of this data? Are these model simulated results? Or actual/real estimates?

Response

Apologies for the confusion, these are simulated irrigation rates as derived from the (IRR) ensemble mean. We updated the caption to clarify this point (note that this now figure S5 due to removal of one figure):

Simulated present-day irrigation rates Q_{irr} as a function of... The binning was performed on the individual IRR ensemble members and bin statistics were subsequently averaged over all ensemble members.

Reviewer 3 Comment 12

SI Line 37: 'Interestingly, (i) CESM has less sensitivity compared to CRU' I do not understand. I see the opposite in Figure S11. Values for CESM are mostly higher and more variable. Furthermore, could you please elaborate on why you think the sensitivity reported in Figure S11 is so much higher and variable in CESM than in CRU v4.02.

Response

The sensitivity we refer to here is the sensitivity to the search window size, that is, the change in values (hence color saturation) along the y-axis in figures S11 and S12 (note: figure numbers are for the previous version; the original figure s12 was omitted and figure s11 became figure s10 in new version of the manuscript). In other words, the more horizontal the dashed lines, the more sensitive to results to search window size. In this context we advanced that CESM results are more sensitive to search window size compared to CRU. But we agree that this difference is only marginal and that emphasizing it may cause confusion, hence we omitted this sentence from the SI.

References

- Bright, R. M., Davin, E., O'Halloran, T., Pongratz, J., Zhao, K., and Cescatti, A. (2017). Local temperature response to land cover and management change driven by non-radiative processes. *Nature Climate Change*, 7(4):296–302.
- Chen, X. and Jeong, S.-J. (2018). Irrigation enhances local warming with greater nocturnal warming effects than daytime cooling effects. *Environmental Research Letters*, 13:024005.
- Cook, B. I., Shukla, S. P., Puma, M. J., and Nazarenko, L. S. (2015). Irrigation as an historical climate forcing. *Climate Dynamics*, 44:1715–1730.
- de Vrese, P., Hagemann, S., and Claussen, M. (2016). Asian irrigation, African rain: Remote impacts of irrigation. *Geophysical Research Letters*, 43(8):3737–3745.
- Demuzere, M., Coutts, A., Göhler, M., Broadbent, A., Wouters, H., van Lipzig, N., and Gebert, L. (2014). The implementation of biofiltration systems, rainwater tanks and urban irrigation in a single-layer urban canopy model. *Urban Climate*, 10:148–170.
- Devanand, A., Huang, M., Ashfaq, M., Barik, B., and Ghosh, S. (2019). Choice of Irrigation Water Management Practice affects Indian Summer Monsoon Rainfall and its Extremes. *Geophysical Research Letters*, in press.
- Duveiller, G., Hooker, J., and Cescatti, A. (2018). The mark of vegetation change on Earth's surface energy balance. *Nature Communications*, 9(1):679.
- Fischer, E. M. and Knutti, R. (2015). Anthropogenic contribution to global occurrence of heavy-precipitation and high-temperature extremes. *Nature Climate Change*, 5:560–565.

- Hooker, J., Duveiller, G., and Cescatti, A. (2018). Data descriptor: A global dataset of air temperature derived from satellite remote sensing and weather stations. *Scientific Data*, 5:1–11.
- Jägermeyr, J., Gerten, D., Heinke, J., Schaphoff, S., Kummu, M., and Lucht, W. (2015). Water savings potentials of irrigation systems: global simulation of processes and linkages. *Hydrology and Earth System Sciences*, 19(7):3073–3091.
- Jones, P. W. (1999). First- and Second-Order Conservative Remapping Schemes for Grids in Spherical Coordinates. *Monthly Weather Review*, 127(9):2204–2210.
- Lawrence, D. M., Oleson, K. W., Flanner, M. G., Thornton, P. E., Swenson, S. C., Lawrence, P. J., Zeng, X., Yang, Z.-L., Levis, S., Sakaguchi, K., Bonan, G. B., and Slater, A. G. (2011). Parameterization improvements and functional and structural advances in Version 4 of the Community Land Model. *Journal of Advances in Modeling Earth Systems*, 3:M03001.
- Lawston, P. M., Santanello, J. A., Zaitchik, B. F., and Rodell, M. (2015). Impact of Irrigation Methods on Land Surface Model Spinup and Initialization of WRF Forecasts. *Journal of Hydrometeorology*, 16:1135–1154.
- Leng, G., Huang, M., Tang, Q., and Lueng, R. (2015). A modeling study of irrigation effects on global surface water and groundwater resources under a changing climate. *Journal of Advances in Modeling Earth Systems*, 7:1285–1304.
- Leng, G., Leung, L. R., and Huang, M. (2017). Significant impacts of irrigation water sources and methods on modeling irrigation effects in the ACME Land Model. *Journal of Advances in Modeling Earth Systems*, 9:1665–1683.
- Liu, T., Yu, L., and Zhang, S. (2019a). Impacts of Wetland Reclamation and Paddy Field Expansion on Observed Local Temperature Trends in the Sanjiang Plain of China. *Journal of Geophysical Research Earth Surface*, 124:414–426.
- Liu, T., Yu, L., and Zhang, S. (2019b). Land Surface Temperature Response to Irrigated Paddy Field Expansion: a Case Study of Semi-arid Western Jilin Province, China. *Scientific Reports*, 9(1):5278.
- Masutomi, Y., Ono, K., Mano, M., Maruyama, A., and Miyata, A. (2016). A land surface model combined with a crop growth model for paddy rice (MATCRO-Rice v.1) - Part 1: Model description. *Geoscientific Model Development*, 9:4133–4154.
- Thiery, W., Davin, E. L., Lawrence, D. M., Hirsch, A. L., Hauser, M., and Seneviratne, S. I. (2017). Present-day irrigation mitigates heat extremes. *Journal of Geophysical Research Atmospheres*, 122:1403–1422.
- Wan, Z. (2008). New refinements and validation of the MODIS Land-Surface Temperature/Emissivity products. *Remote Sensing of Environment*, 112(1):59–74.
- Winckler, J., Reick, C. H., Luyssaert, S., Cescatti, A., Stoy, P. C., Lejeune, Q., Raddatz, T., Chlond, A., Heidkamp, M., and Pongratz, J. (2019). Different response of surface temperature and air temperature to deforestation in climate models. *Earth System Dynamics Discussions*, October:1–17.

-
- Xie, X. and Cui, Y. (2011). Development and test of SWAT for modeling hydrological processes in irrigation districts with paddy rice. *Journal of Hydrology*, 396(1):61–71.

REVIEWERS' COMMENTS:

Reviewer #2 (Remarks to the Author):

Overall, I believe that the authors have addressed most (if not all) comments sufficiently; they added the proposed analysis from reviewer #3 and replied to all comments in - from my perspective - sufficient detail and explanation. However, admittedly, I was not able to follow the full discussion on the comparison of the results as presented in the manuscript and the land surface temperature analysis from MODIS, as suggested by reviewer #3. I did, however, grasp the intention of the reviewer's comment and the context of the analysis. Therefore, I took a step back and looked at the bigger picture, which left one main concern, addressed by comments from both reviewers: the uncertainty and credibility of the results due to (i) the employed irrigation technique (reviewer #1) along with (ii) observational uncertainty and methodological choices (reviewer #3).

So, for me, the main question to address this concern really is:

How would the results change if

- other observations (incl. observational uncertainty),
 - other (slightly different) time periods, or
 - other models, or
 - other implementations of irrigation in these models, or
 - modified analysis methods (mixing / unmixing window technique),
- were used?

Reconciling the results and discussion, it seems like changes can only be expected in the strength of the signal and the accompanied likelihood ratio. However, the overall signal and message - an irrigation induced cooling effect evident in the present century - would likely remain the same (as illustrated over different time periods and with different observations). Considering this, the only criticism that remains not fully answered is the acknowledgement of this uncertainty in the manuscript itself. However, the methodology is well described and reproducible (code and data have been published), and the authors elaborate on the assumptions and limits inherent in the methodology, showing further results in the Supplementary. All main results are furthermore illustrated as the mean of an ensemble and expressed in a probabilistic terminology rather than absolute terms.

Personally, I believe that the study represents a novel finding - even if restricted by common assumptions and the use of (uncertain) observations and data sets. As a consequence, I believe that the manuscript in its current state portrays novel findings, which are worth publishing.

For further details, I summarize how the authors have addressed the two main comments from reviewers #1 and #3 and expand on my personal reasoning.

-- Implementation of irrigation techniques in CESM

+ The main criticism of reviewer #1 is that the applied irrigation technique is a simplified version of reality, especially in South Asia;

+ The authors acknowledge this shortcoming and expand on the effects of this simplified representation in the manuscript.

+ Personally, I think that (i) the study constitutes an advancement over previous studies even if irrigation is incorporated in a simplified manner (and when is a simulation ever a 'perfect' replica of reality?); and (ii) that the shortcomings of this simplified representation are discussed in detail in both the response as well as the manuscript. Furthermore, I agree with the author's statement

that the implementation of different irrigation techniques cannot be tackled within a review (at least not properly) but is rather the work of a few months to years.

-- Affirmation of the credibility

+ The main criticism of reviewer #3 is a lack of credibility w.r.t. the CESM simulations along with observational uncertainty.

+ The authors performed the analysis as proposed by reviewer #3 and corrected the manuscript accordingly. They refer to earlier studies (Thiery et al., 2017) for a comprehensive validation of the same simulations and expand on this in response to a comment from reviewer #1.

+ Personally, I believe that the probabilistic evaluation, based on ensemble simulations and confirmed by a set of observations, confirms the credibility of the results. The only criticism that could remain unaddressed (even though reviewer #3 did not directly point to this) is that the authors do not elaborate on other uncertainties inherent in the results, despite explicitly highlighting these in the response ("For us, the large difference between both versions of the same observational data set is a clear sign that the observational data set is characterised by uncertainty, just like the model simulations are. This is confirmed by the fact that the main difference between both CRU versions is a new interpolation algorithm, not an increase in the number of stations included in the data product. Moreover, the new analysis on CESM and MODIS land surface temperatures (see below) suggests that CESM underestimates rather than overestimates the MODIS-based cooling estimate, confirming that both models and observational products need to be treated with caution here."). If the authors decided not to use multiple observations in the manuscript to highlight this uncertainty accordingly, this could be mentioned. However, in that case, a single sentence would suffice, in my opinion.

Reviewer #4 (Remarks to the Author):

I have read through the paper, and the responses to the reviews. I am satisfied with the author's responses and I think this is a solid contribution. However, I have a few minor comments that in my opinion should be discussed in the text.

1) From my point of view, the cooling role of irrigation is very local. It may for example lower the maximum temperature by 5 degrees locally and almost no effect over the surroundings. This cannot be estimated in the present work since inside the grid-cell both of irrigated and non-irrigated areas share the same climate (2m air temperature). Even if the average temperature over the grid-cell may be cooled by irrigation, this does not mean that the population that is mainly living far from the irrigated areas will feel such a cooling (or less warming) effect. I think that the number of 0.79-1.29 billion of people is largely overestimated.

2) The direct evaporation from rivers, dams and inundated areas is not simulated by the model. This may increase the evaporation contrast between irrigated and non-irrigated areas. Thus, the used method may overestimate the irrigation effect. On the other hand, as mentioned by reviewer 3 comment 8, using a prescribed seasonal LAI in the model may result in a mismatch with reality. I am not fully convinced by the answer to that comment. This is important as LAI may damp the extreme climate impacts by modulating the surface albedo and latent heat fluxes. In case of drought, for instance, while irrigated crops may not be affected much (as one would assume irrigation is sufficient to ensure proper 'normal' growth of LAI), the non-irrigated crops around that serve as comparison would be wrongly modelled as having a high LAI, lower albedo and warmer extremes than should be. This also leads to an overestimation of the cooling role of irrigation by the CESM model.

3) The authors used a multiple linear regression model in order to estimate the irrigation effects. While, the real world is more complicated than just being represented by multiple linear regression. I am not for redoing a more complicated analysis but adding a few sentences

explaining why using a multilinear regression and why only lat, lon and elevation are used as predictors could help the readers. Why, for example, not using the distance to the sea (ocean) as predictor.

4) From what I understood, CRU3.22 and CRU4.2 leads to a different result. As I see it, the main difference may come from the 1901-1930 period where the number of in-situ datasets are very low. In my opinion using another version may lead to an opposite signal. I understand the point of view of the reviewer 3 who suggested keeping only the last version CRU4.2, but I recommend to add at least one sentence about the robustness of the CRU analysis. Note that, CRU is based on in-situ observations that measures temperature on non-irrigated areas. They may be close but not inside which makes hard to extract the temperature related to the irrigation.

Warming of hot extremes alleviated by expanding irrigation

Submitted to Nature Communications

November 15, 2019

Wim Thiery^{1, 2}, Auke J. Visser³, Erich M. Fischer¹, Mathias Hauser¹, Annette L. Hirsch⁴,
David M. Lawrence⁵, Quentin Lejeune⁶, Edouard L. Davin¹, Sonia I. Seneviratne¹
wim.thiery@env.ethz.ch

¹ETH Zurich, Institute for Atmospheric and Climate Science, Universitaetsstrasse 16, 8092 Zurich, Switzerland.

²Vrije Universiteit Brussel, Department of Hydrology and Hydraulic Engineering, Pleinlaan 2, 1050 Brussels, Belgium.

³Wageningen University, Meteorology and Air Quality group, Droevendaalsesteeg 3a, 6708PB, Wageningen, the Netherlands.

⁴University of New South Wales, ARC Centre of Excellence for Climate Extremes, 2052 Sydney, Australia.

⁵National Center for Atmospheric Research, Boulder, Colorado, USA.

⁶Climate Analytics, Ritterstrasse 3, 10969 Berlin, Germany.

Contents

1	Reviewer 1	3
2	Reviewer 2	3
	Reviewer 2 Comment 1	3
	Reviewer 2 Comment 2	3
3	Reviewer 3	5
4	Reviewer 4	5
	Reviewer 4 Comment 1	5
	Reviewer 4 Comment 2	5
	Reviewer 4 Comment 3	6
	Reviewer 4 Comment 4	7
	Reviewer 4 Comment 5	8

Abstract

We would like to once again thank all reviewers for their dedicated time reviewing the response letter and revised manuscript, and for their useful and constructive sug-

gestions in this new round. We carefully addressed all comments by the reviewers. In addition, we incorporated all suggestions by the Editor regarding the manuscript formatting.

1 Reviewer 1

2 Reviewer 2

Reviewer 2 Comment 1

Overall, I believe that the authors have addressed most (if not all) comments sufficiently; they added the proposed analysis from reviewer 3 and replied to all comments in - from my perspective - sufficient detail and explanation. However, admittedly, I was not able to follow the full discussion on the comparison of the results as presented in the manuscript and the land surface temperature analysis from MODIS, as suggested by reviewer 3. I did, however, grasp the intention of the reviewer's comment and the context of the analysis. Therefore, I took a step back and looked at the bigger picture, which left one main concern, addressed by comments from both reviewers: the uncertainty and credibility of the results due to (i) the employed irrigation technique (reviewer 1) along with (ii) observational uncertainty and methodological choices (reviewer 3).

So, for me, the main question to address this concern really is: How would the results change if - other observations (incl. observational uncertainty), - other (slightly different) time periods, or - other models, or - other implementations of irrigation in these models, or - modified analysis methods (mixing / unmixing window technique), were used?

Reconciling the results and discussion, it seems like changes can only be expected in the strength of the signal and the accompanied likelihood ratio. However, the overall signal and message - an irrigation induced cooling effect evident in the present century - would likely remain the same (as illustrated over different time periods and with different observations). Considering this, the only criticism that remains not fully answered is the acknowledgement of this uncertainty in the manuscript itself. However, the methodology is well described and reproducible (code and data have been published), and the authors elaborate on the assumptions and limits inherent in the methodology, showing further results in the Supplementary. All main results are furthermore illustrated as the mean of an ensemble and expressed in a probabilistic terminology rather than absolute terms.

Personally, I believe that the study represents a novel finding - even if restricted by common assumptions and the use of (uncertain) observations and data sets. As a consequence, I believe that the manuscript in its current state portrays novel findings, which are worth publishing.

Response

We thank Reviewer 2 for their time revising the manuscript and response letter, and for the recommendation to publish this study.

Reviewer 2 Comment 2

Personally, I believe that the probabilistic evaluation, based on ensemble simulations and confirmed by a set of observations, confirms the credibility of the results. The only criticism that could remain unaddressed (even though reviewer 3 did not directly point to this) is that the authors do not elaborate on other uncertainties inherent in the results, despite explicitly highlighting these in the response ('For us, the large difference between both versions of the same observational data set is a clear sign that the observational data set is characterised by uncertainty, just like the model simulations are. This is confirmed by the fact that the main difference between both CRU versions is a new interpolation algorithm, not an increase in the number of stations included in the data product. Moreover, the new analysis on CESM and MODIS land surface temperatures (see below) suggests that CESM underestimates rather than overestimates the MODIS-based cooling estimate, confirming that both models and observational products need to be treated with caution here.'). If the authors decided not to use multiple observations in the manuscript to highlight this uncertainty accordingly, this could be mentioned. However, in that case, a single sentence would suffice, in my opinion.

Response

Following the recommendation from Reviewer 3 we indeed omitted all references to the older CRU version. Following the suggestion from Reviewer 2, we now added a short paragraph in the Methods section to highlight the uncertainty of the observational product used for the analysis:

Finally, we also applied our analysis to an earlier version of the temperature data set (CRU TS v3.22). The main difference between both versions is a new interpolation algorithm, not an increase in the number of stations included in the data product. The results based on the earlier CRU version show a substantially stronger irrigation-induced cooling signal, highlighting that, next to the model simulations, also the observation-based estimates are characterised by uncertainty and thus need to be treated with care.

3 Reviewer 3

4 Reviewer 4

Reviewer 4 Comment 1

I have read through the paper, and the responses to the reviews. I am satisfied with the author's responses and I think this is a solid contribution. However, I have a few minor comments that in my opinion should be discussed in the text.

Response

We thank the Reviewer for the appreciation of the manuscript, the response letter and revisions we implemented in the original manuscript. Below, we address the issues raised in this new review round.

Reviewer 4 Comment 2

From my point of view, the cooling role of irrigation is very local. It may for example lower the maximum temperature by 5 degrees locally and almost no effect over the surroundings. This cannot be estimated in the preset work since inside the grid-cell both of irrigated and no irrigated areas share the same climate (2m air temperature). Even if the average temperature over the grid-cell may be cooled by irrigation, this does not mean that the population that is mainly living far from the irrigated areas will feel such a cooling (or less warming) effect. I think that the number of 0.79-1.29 billion of peoples is largely overestimated.

Response

We agree with the assessment of the Reviewer regarding the local effect of irrigation. In fact, we demonstrated this in our previous study (Thiery et al., 2017b), by comparing the grid-scale surface temperature response to irrigation against the subgrid-scale response (Illustration 1a,c). The same conclusion applies to evapotranspiration, whereby we even showed that the added moisture in the boundary layer triggers a negative feedback within the grid cell with less evapotranspiration from the other subgrid tiles (Illustration 1b,d). Unfortunately, as correctly noted by the Reviewer, it is not possible to quantify this subgrid-scale effect for the 2m air temperature, as this variable is influenced by the surface energy balance as well as boundary layer characteristics (which are only computed at grid scale).

Regarding the number of people exposed to irrigation-induced cooling, we would like to emphasize that we already take this concern of subgrid-scale population density heterogeneity into account. In particular, we have two estimates for number of people exposed to irrigation-induced cooling: one where we apply the grid-scale cooling to the grid cell total population (**upper bound**, as some people live away from the irrigated crops), and one where we apply the irrigation-induced cooling only to the rural population living in that grid cell (**lower bound**, as the local cooling where the rural population lives will be stronger than the grid-scale average). This is currently explained in the method section as follows:

[redacted]

While our upper estimate assumes that all people living within a $0.90^\circ \times 1.25^\circ$ grid cell are equally exposed to the grid cell mean PR , our lower bounds neglects that the rural irrigation signature will much stronger and thus widespread (Thiery et al., 2017a).

Reviewer 4 Comment 3

The direct evaporation from rivers, dams and inundated areas is not simulated by the model. This may increase the evaporation contrast between irrigated and non-irrigated areas. Thus, the used method may overestimate the irrigation effect. On the other hand, as mentioned by reviewer 3 comment 8, using a prescribed seasonal LAI in the model may result in a mismatch with reality. I am not fully convinced by the answer to that comment. This is important as LAI may damp the extreme climate impacts by modulating the surface albedo and latent heat fluxes. In case of drought, for instance, while irrigated crops may not be affected much (as one would assume irrigation is sufficient to ensure proper 'normal' growth of LAI), the non-irrigated crops around that serve as comparison would be wrongly modelled as having a high LAI, lower albedo and warmer extremes than should be. This also leads to an overestimation of the cooling role of irrigation by the CESM model.

Response

We would like to note that the model does simulate evaporation from rivers, natural lakes and reservoirs (Oleson et al., 2013). Open water in every pixel is prescribed according following a MODIS-based data set (Lawrence and Chase, 2007), and the lake model (Subin et al., 2012) is activated for this open water fraction (one could argue that a lake model is not the best way to represent evaporation from rivers, but our ongoing research show that the grid scale fraction of rivers is small compared to that of lakes and reservoirs so this will have limited influence on the resulting open water evaporation flux).

Regarding LAI, we note that the model prescribes a climatological LAI in CLM4-SP mode. Hence, vegetation responses to drought are not directly considered. We are very much aware that this is a limitation of the model, but underline that irrigation and interactive phenology are mutually exclusive in the model version we used for this study. In the new CESM version, which was recently released, this issue is solved. Hence we believe that future irrigation impact studies will have a more realistic representation of LAI responses to droughts and irrigation.

Reviewer 4 Comment 4

The authors used a multiple linear regression model in order to estimate the irrigation effects. While, the real world is more complicated than just being represented by multiple linear regression. I am not for redoing a more complicated analysis but adding a few sentences explaining why using a multilinear regression and why only lat, lon and elevation are used as predictors could help the readers. Why, for example, not using the distance to the sea (ocean) as predictor.

Response

Lejeune et al. (2018) extensively tested for which predictors are appropriate to quantify the effects of deforestation on temperature extremes. In this analysis they were aided by the availability of multi-model factorial experiments (simulations with and without deforestation) against which the results of their window-searching method could be compared. Their analysis resulted in a list of four predictors including deforestation, latitude, longitude and elevation.

In the current study we tested for including deforestation next to irrigation (i.e. adding irrigation to the set of explanatory variables used by Lejeune et al. (2018), but we found that including this variable did not increase the explanatory power of the overall regression model. Hence we decided to omit this variable.

We now added some text to the method section to clarify this point:

The selection of the three spatial predictors next to irrigation expansion was informed by earlier tests (Lejeune et al., 2018) showing that inclusion of these factors succeeds in filtering out the most important natural climate gradients within the search window.

Reviewer 4 Comment 5

From what I understood, CRU3.22 and CRU4.2 leads to a different result. As I see it, the main difference may come from the 1901-1930 period where the number of in-situ datasets are very low. In my opinion using another version may lead to an opposite signal. I understand the point of view of the reviewer 3 who suggested keeping only the last version CRU4.2, but I recommend to add at least one sentence about the robustness of the CRU analysis. Note that, CRU is based on in-situ observations that measures temperature on non-irrigated areas. They may be close but not inside which makes hard to extract the temperature related to the irrigation.

Response

Thank you for this suggestion, which was also made Reviewer 3 in this round of reviews. We have now added a paragraph in the method section describing this issue:

Finally, we also applied our analysis to an earlier version of the temperature data set (CRU TS v3.22). The main difference between both versions is a new interpolation algorithm, not an increase in the number of stations included in the data product. The results based on the earlier CRU version show a substantially stronger irrigation-induced cooling signal, highlighting that, next to the model simulations, also the observation-based estimates are characterised by uncertainty and thus need to be treated with care.

References

- Lawrence, P. J. and Chase, T. N. (2007). Representing a new MODIS consistent land surface in the Community Land Model (CLM 3.0). *Journal of Geophysical Research*, 112:G01023.
- Lejeune, Q., Davin, E. L., Gudmundsson, L., Winckler, J., and Seneviratne, S. I. (2018). Historical deforestation locally increased the intensity of hot days in northern mid-latitudes. *Nature Climate Change*, 8(5):386–390.
- Oleson, K. W., Lawrence, D. M., Bonan, G. B., Drewniak, B., Huang, M., Charles, D., Levis, S., Li, F., Riley, W. J., Zachary, M., Swenson, S. C., Thornton, P. E., Bozbiyik, A., Fisher, R., Heald, C. L., Kluzek, E., Lamarque, F., Lawrence, P. J., Leung, L. R., Muszala, S., Ricciuto, D. M., and Sacks, W. (2013). Technical description of version 4.5 of the Community Land Model (CLM). *NCAR Technical Note*, 1:420.
- Subin, Z. M., Riley, W. J., and Mironov, D. (2012). An improved lake model for climate simulations: Model structure, evaluation, and sensitivity analyses in CESM1. *Journal of Advances in Modeling Earth Systems*, 4:M02001.
- Thiery, W., Davin, E. L., Lawrence, D. M., Hirsch, A. L., Hauser, M., and Seneviratne, S. I. (2017a). Present-day irrigation mitigates heat extremes. *Journal of Geophysical Research Atmospheres*, 122:1403–1422.

Thiery, W., Gudmundsson, L., Bedka, K., Semazzi, F., Lhermitte, S., Willems, P., Van Lipzig, N., and Seneviratne, S. (2017b). Early warnings of hazardous thunderstorms over Lake Victoria. *Environmental Research Letters*, 12(7).